# Learning to Theorize the World from Observation

Doojin Baek [* 1]   Gyubin Lee [* 1]   Junyeob Baek [1]   Hosung Lee [1]   Sungjin Ahn [1]

## Abstract

What does it mean to understand the world? Contemporary world models often operationalize understanding as accurate future prediction in latent or observation space. Developmental cognitive science, however, suggests a different view: human understanding emerges through the construction of internal theories of how the world works, even before mature language is acquired. Inspired by this theory-building view of cognition, we introduce *Learning-to-Theorize*, a learning paradigm for inferring explicit explanatory theories of the world from raw, non-textual observations. We instantiate this paradigm with the *Neural Theorizer (NEO)*, a probabilistic neural model that induces latent programs as a learned Language of Thought and executes them through a shared transition model. In NEO, a theory is represented as an executable, compositional program whose learned primitives can be systematically recombined to explain novel phenomena. Experiments show that this formulation enables explanation-driven generalization, allowing observations to be understood in terms of the programs that generate them.

## 1. Introduction

*"Instead of trying to produce a programme to simulate the adult mind, why not rather try to produce one which simulates the child's?"*            (Alan Turing, 1950)

In his seminal 1950 paper (Turing, 1950), Alan Turing suggested that artificial intelligence (AI) should not aim first to reproduce the fully developed adult mind, but instead to understand how an intelligent mind could be learned from its earliest stages. This child-centered view reframes the goal of AI: rather than asking only how to imitate mature intelligent behavior, it asks what kind of learning process gives rise to such behavior in the first place.

A central insight from developmental cognitive science is that children are not merely passive predictors of sensory inputs or imitators of linguistic behavior. Long before acquiring mature language, they appear to construct and revise internal theories of how the world works (Goddu & Gopnik, 2024; Dragoi, 2024; Liang et al., 2025; Luettgau et al., 2025). This view, often referred to as the *theory-theory* of cognitive development or the *baby as a scientist* perspective (Gopnik et al., 1999), holds that early cognition is guided by the discovery of structured, reusable, and compositional explanations of observed phenomena (Goddu & Gopnik, 2024; Dautriche & Chemla, 2025). In this sense, the child's mind is not simply a machine for forecasting what will happen next, but a *theory-building system* that learns to explain *how* and *why* the world changes.

This theory-building view of cognition points to a richer notion of understanding: the construction of internal *explanatory structures* that capture how observations are generated and transformed. These structures should be reusable across instances, compositional across novel situations, and explicit enough to support intervention and counterfactual reasoning (Lake & Baroni, 2023; Schölkopf et al., 2021). This implies a learning objective beyond future prediction: inferring the abstract mechanisms that explain them.

However, most contemporary AI systems, including recent world models, are not designed around this objective. They are primarily optimized for future prediction in latent or observation space, reconstruction quality, or task-specific performance (Maes et al., 2026; Zhu et al., 2024; Hafner et al., 2019). These objectives do not require models to discover explicit, reusable mechanisms that explain how observations are generated and transformed. Instead, they can often be satisfied by learning entangled composite transformations that capture correlations among observed inputs and outputs. Such representations lack the compositional structure needed to systematically recombine familiar components in unfamiliar ways, making them brittle under distribution shift or compositional generalization tests (Chollet, 2019).

This gap highlights a central open problem: *Can an artificial system learn to construct explicit explanatory theories of the world merely by observing raw, non-textual sensory*

---

*Equal contribution [1]KAIST. Correspondence to: Sungjin Ahn <sungjin.ahn@kaist.ac.kr>.

*Proceedings of the 43rd International Conference on Machine Learning*, Seoul, South Korea. PMLR 306, 2026. Copyright 2026 by the author(s).

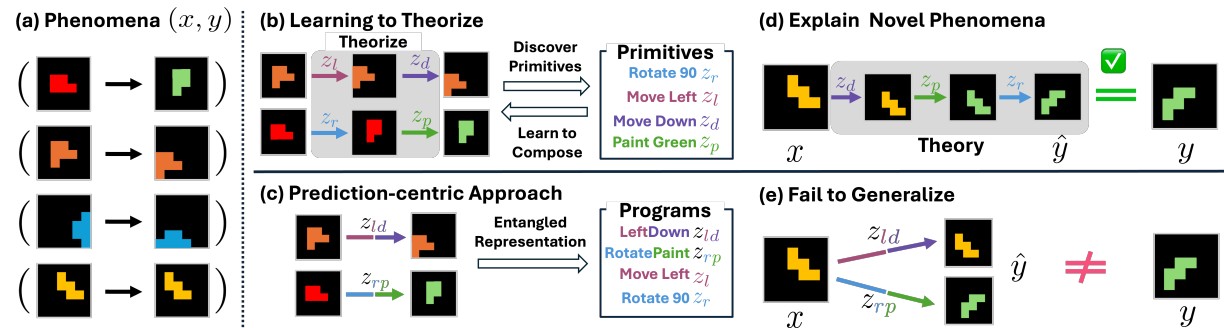

*Figure 1.* **Learning to Theorize (L2T) Framework. (a)** Training data consists of observation pairs $(x, y)$ generated by unobserved true programs. **(b)** Under L2T, the model learns to discover reusable primitives (Rotate, Left, Down, and Paint) and to compose them into executable theories. **(c)** Without L2T, the model instead memorizes entangled composite primitives (e.g., Left-Down) as indecomposed single units. **(d)** Once the model has learned to theorize, novel phenomena (e.g., Down-Paint-Rotate) can be explained by recombining learned primitives. **(e)** In contrast, memorized entangled representations fail to generalize to unseen programs.

*inputs?* Addressing this problem requires a shift in learning objectives: from fitting input–output mappings to discovering structured, compositional mechanisms that explain how observations are generated and transformed.

In this paper, we take a step toward this goal by introducing *Learning-to-Theorize (L2T)*, a learning paradigm for inducing explicit theories from observation alone. Rather than learning task-specific predictors or policies, L2T aims to infer *programmatic explanations* of how observations are generated and transformed. A theory is represented as an executable, compositional program whose learned primitives can be systematically recombined to explain novel phenomena. Learning therefore targets reusable explanatory structures—internal "mental programs" (Chater & Oaksford, 2013; Dehaene et al., 2022).

To instantiate this paradigm, we propose the *Neural Theorizer (NEO)*, a probabilistic neural model for inferring latent executable programs from paired observations. Given an observation pair $(x, y)$, NEO induces a discrete program as a learned Language of Thought (Fodor, 1975), and executes it through a shared transition model to reconstruct the target observation $y$. By sharing primitive operations and the executor across examples, NEO is encouraged to discover reusable operations, compose them into structured programs, and infer unseen compositions at test time, including programs longer than those observed during training.

Unlike prior approaches that rely on symbolic supervision (Nye et al., 2020), task grouping (Chollet, 2019), or explicit program annotations (Mao et al., 2019), NEO learns solely from raw observation pairs. This makes it possible to train on minimally curated observational data, such as temporally separated frames from trajectories, without assuming language descriptions, task labels, or ground-truth programs. At test time, NEO is evaluated not only by how accurately it reconstructs target observations, but also by whether its inferred programs transfer across distinct instances generated by the same underlying mechanism. This

directly tests whether the model has learned a reusable theory rather than an instance-specific mapping.

Our contributions are summarized as follows:

- We formulate **Learning-to-Theorize (L2T)** as a learning paradigm for inferring executable theories from raw observations, without relying on language, task labels, or program supervision.
- We propose **NEO**, a probabilistic neural model that learns a latent Language of Thought and induces compositional executable programs as explanations of how observations are generated and transformed.
- We introduce the **Observation-to-Theory Induction Benchmark (OTIB)**, a benchmark for evaluating whether models can infer reusable theories from observation without program supervision or task grouping.
- We demonstrate that NEO achieves explanation-driven generalization by transferring inferred programs across instances, recombining primitives into unseen program compositions, and generalizing to programs longer than those observed during training.

## 2. Learning to Theorize

We define the ability to theorize the world as the capacity to (i) *discover reusable abstract primitives* across phenomena, (ii) *learn how to compose* them into structured explanations of complex observations, and (iii) *explain novel phenomena* by forming new compositions of the same primitives. While theories may be instantiated in various concrete forms (e.g., natural language, probabilistic programs, or symbolic programs), we formulate *Learning-to-Theorize (L2T)* as a problem of latent neural program induction from observation, in which programs act as executable representations of theories. Accordingly, we use the terms *theory* and *program* interchangeably when no confusion arises. An overview of the L2T framework is shown in Figure 1.

**Phenomenon and Generative Process.** We model a *phe-*

*nomenon* as a pair of observations $(x, y)$, where $x \sim p(x)$ is a source observation and $y$ is a corresponding target observation. For example, $x$ may represent an observation of an apple hanging from a tree, while $y$ represents an observation of the apple after it has fallen to the ground. We assume that each phenomenon is generated by an underlying but unobserved program (or causal mechanism) $\tau$ that transforms $x$ to $y$, i.e., $p(y \mid x, \tau)$.

**Compositional Structure of Programs.** A *program* is a compositional object formed by combining a finite set of primitive operations, whose execution defines a structured transformation of observations. Formally, let $\mathcal{Z} = \{z_1, z_2, \ldots, z_M\}$ denote a set of primitive operations, where each primitive $z_i$ is associated with an execution function $f_{z_i} : \mathcal{X} \to \mathcal{X}$. A program (or theory) $\tau$ of length $K$ is defined as an ordered sequence of primitives,

$$\tau = (z_{i_1}, z_{i_2}, \ldots, z_{i_K}), \qquad z_{i_k} \in \mathcal{Z}.$$

Program execution corresponds to functional composition of the associated primitive functions,

$$f_\tau = f_{z_{i_K}} \circ f_{z_{i_{K-1}}} \circ \cdots \circ f_{z_{i_1}}.$$

Given a source observation $x$, the target observation is obtained by executing the program $y = f_\tau(x)$, or more generally, by a conditional distribution $p(y \mid x, \tau) = p\big(y \mid f_\tau(x)\big)$. This formulation highlights that programs represent compositional and reusable transformations of observations.

**Compositional Generalization and Training Dataset.** We denote by $\mathcal{Z}^*$ the space of all finite-length sequences of primitives in $\mathcal{Z}$. Because the compositional program space $\mathcal{Z}^*$ grows exponentially with program length, only a small subset of all possible programs can be realized during training. We therefore assume that training data are generated by a restricted subset $\mathcal{T}_{\text{train}} \subset \mathcal{Z}_K^*$, where $\mathcal{Z}_K^* \subset \mathcal{Z}^*$ denotes the set of programs of length at most $K$.

The training dataset $\mathcal{D}_{\text{train}} = \{(x_n, y_n)\}_{n=1}^N$ consists of phenomena generated as $y_n = f_{\tau_n}(x_n)$ with $\tau_n \sim p_{\text{train}}(\tau)$ and $\tau_n \in \mathcal{T}_{\text{train}}$. Importantly, the programs $\{\tau_n\}$ and their associated functions $\{f_{\tau_n}\}$ are latent and never observed; only the resulting observation pairs $(x_n, y_n)$ are available for learning. Consequently, training seeks to jointly infer a theory $\tau$ for each phenomenon and to learn a shared set of execution functions that realize these theories.

This dataset assumption is highly general and requires minimal task-specific curation. In a canonical world-modeling setting, one may simply set $x_n = x_{t_n}$ and $y_n = x_{t_n + t'_n}$ as temporally separated observations, where both the time index $t_n$ and the time lag $t'_n$ are unobserved. Unlike datasets such as ARC-AGI (Chollet, 2019), which assume few-shot groups of examples sharing the same underlying program, our formulation imposes no such structure: $\mathcal{D}_{\text{train}}$ consists of

i.i.d. phenomena. Thus, the approach is directly applicable to large-scale datasets.

**Test-Time Generalization.** At test time, we evaluate the model on phenomena generated by programs drawn from a disjoint subset $\mathcal{T}_{\text{test}}$ satisfying $\mathcal{T}_{\text{test}} \subset \mathcal{Z}_{K'}^* \subset \mathcal{Z}^*$, $\mathcal{T}_{\text{test}} \cap \mathcal{T}_{\text{train}} = \varnothing$, and $K' > K$. Thus, test-time evaluation requires not only *compositional generalization* to previously unseen programs, but also *length generalization*, i.e., productivity, to longer compositions than those realized during training.

Given a test phenomenon $(x, y)$, the model must infer a latent program $\hat{\tau}$ that explains the observation by composing previously learned primitives, $\hat{\tau} = \arg\max_{\tau \in \mathcal{T}_{\text{test}}} p(\tau \mid x, y)$, with $\hat{\tau} \notin \mathcal{T}_{\text{train}}$. We interpret successful inference in this regime as evidence of *Learning-to-Theorize*.

**Evaluation: Program Transferability.** We evaluate induced theories at the execution level by testing whether inferred programs act as reusable and transferable compositional explanations. At test time, we consider pairs of phenomena $(x^{(1)}, y^{(1)})$ and $(x^{(2)}, y^{(2)})$ generated by the same latent program $\tau \in \mathcal{T}_{\text{test}}$. The model first infers a program $\hat{\tau}$ from $(x^{(1)}, y^{(1)})$ and then applies it to $x^{(2)}$ to obtain $\hat{y}^{(2)} = D_\theta(f_{\hat{\tau}}(x^{(2)}))$. Performance is measured by an observation-space error $d_{\text{obs}}(\hat{y}^{(2)}, y^{(2)})$. This protocol assesses whether the learned theory captures a transferable generative mechanism, rather than merely fitting individual input–output pairs.

## 3. Neural Theorizer

To address the L2T problem, we propose the Neural Theorizer (NEO), a probabilistic neural architecture that learns to infer latent executable programs as theories from paired observations; Figure 2 provides an overview of the architecture.

### 3.1. Probabilistic Modeling

NEO is trained to maximize the conditional likelihood $p_\theta(y \mid x)$. To explicitly model theory construction, we introduce two latent variables: a program $\tau = (z_{i_1}, \ldots, z_{i_K})$ and its execution trace $s = (s_1, \ldots, s_{K+1})$. Under a Markov assumption, the conditional distribution is written as

$$p_\theta(y \mid x) = \int p_\theta(y \mid s_{K+1}) \, p_\theta(\tau, s \mid x) \, d\tau \, ds. \quad (1)$$

The prior over programs and execution traces factorizes according to the following generative process: $p_\theta(\tau, s \mid x)$

$$= p_\theta(s_1 \mid x) \prod_{k=1}^K p_\theta(z_{i_k} \mid s_k) \, p_\theta(s_{k+1} \mid s_k, z_{i_k}). \quad (2)$$

Here, $p_\theta(s_1 \mid x)$ defines an encoder mapping observations to latent states, $p_\theta(z_{i_k} \mid s_k)$ defines a theory programmer

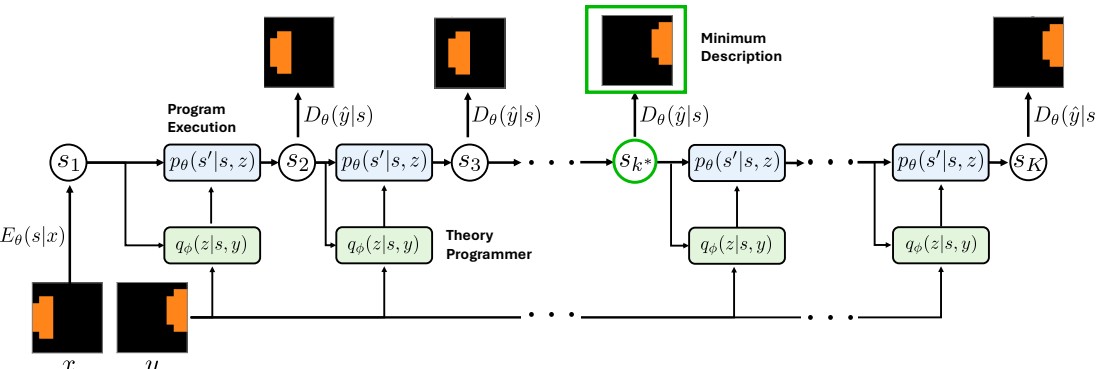

*Figure 2.* **Computation graph of Neural Theorizer (NEO)**. NEO infers a latent program by iteratively selecting a primitive $z_{ik}$ with the theory programmer $q_\phi(z_{ik} \mid s_k, y)$ and executing it via the transition model $p_\theta(s_{k+1} \mid s_k, z_{ik})$. Each intermediate state $s_k$ is decoded into a full reconstruction $\hat{y}_k = D_\theta(s_k)$; through state grounding (Sec. 3.4), these intermediate predictions are explicitly regularized to remain valid observations, preventing degenerate or blurry intermediate states. The MDL criterion selects the shortest accurate explanation length $k^*$ (green), which in turn provides a learning signal that favors short yet accurate program compositions (Sec. 3.3).

that selects primitive operations, and $p_\theta(s_{k+1} \mid s_k, z_{i_k})$ defines a shared Markov transition operator implementing primitive execution. For simplicity, we assume a fixed program length $K$; in Section 3.3, we introduce a method for adaptive program length selection.

Since the marginalization in Eq. (1) is intractable, we introduce a variational posterior $q_\phi(\tau, s \mid x, y)$ to approximate the true posterior and optimize the evidence lower bound (Kingma & Welling, 2013; Jordan et al., 1999):

$$
\begin{aligned}
\log p_\theta(y \mid x) \ \geq \ & \mathbb{E}_{q_\phi(\tau,s|x,y)}[\log p_\theta(y \mid s_{K+1})] \\
& - \mathrm{KL}\big(q_\phi(\tau, s \mid x, y) \,\|\, p_\theta(\tau, s \mid x)\big). \quad (3)
\end{aligned}
$$

### 3.2. Theory Programmer

We define the variational posterior over programs and execution traces as follows: $q_\phi(\tau, s \mid x, y)$

$$
= \underbrace{p_\theta(s_1 \mid x)}_{\text{encoder}} \prod_{k=1}^{K} \underbrace{q_\phi(z_{i_k} \mid s_k, y)}_{\text{theory programmer}} \underbrace{p_\theta(s_{k+1} \mid s_k, z_{i_k})}_{\text{program execution}}, \quad (4)
$$

where the encoder and the execution model are shared with the generative model in Eq. (2).

The *theory programmer* $q_\phi(z_{i_k} \mid s_k, y)$ defines a goal-conditioned policy over primitive operations. Given the current latent state $s_k$ and the target observation $y$, it selects the next primitive $z_{i_k}$ so as to steer the execution trace toward a latent state that explains $y$, thereby inducing a compositional program without explicit program supervision.

Concretely, we implement $q_\phi(z_{i_k} \mid s_k, y)$ as a neural network that takes as input the current latent state $s_k$ and the encoded target observation $s_y = E_\theta(y)$, and outputs a categorical distribution over $M'$ primitive categories. Since the true number of primitives $M$ is unknown, $M'$ is treated as a hyperparameter. In practice, this discrete selection is

realized via a vector-quantized variational autoencoder (VQ-VAE) (Van Den Oord et al., 2017), which provides a discrete codebook $\mathcal{E} = \{e_1, \dots, e_{M'}\}$ of primitive symbols while enabling end-to-end training of the theory programmer.

Importantly, the primitives $\{z_i\}$ form the vocabulary of a learned *Language of Thought*: they are abstract symbols without predefined semantics. Their meaning is not specified *a priori*, but is induced through the shared execution model, which assigns operational semantics by defining how each symbol transforms latent states. In this way, NEO jointly learns both the *syntax* of programs (how primitives are composed) and their *semantics* (how these compositions realize state transitions) directly from observation.

By iteratively applying the theory programmer and the shared execution model, NEO constructs an explicit execution trace $s_1 \rightarrow s_2 \rightarrow \cdots \rightarrow s_{K+1}$ that realizes the inferred program as a sequence of latent state transitions. The final state $s_{K+1}$ is then decoded to reconstruct the target observation $\hat{y}_\tau = D_\theta(s_{K+1})$.

### 3.3. Minimum Description Length Principle

Assuming a fixed program length $K$ is unrealistic, as it forces simple phenomena to be over-decomposed into unnecessarily long programs, producing fine-grained primitives with limited reuse and increasing the risk of overfitting. To address this issue, we adopt the Minimum Description Length (MDL) principle (Grünwald, 2007) and assume that, among competing explanations, the theory that generates the shortest program provides the most compositional and reusable set of primitives and their operations.

Specifically, we favor explanations that achieve both (i) low reconstruction loss and (ii) short program length. Concretely, for each intermediate execution step $k$, we compute

a reconstruction $\hat{y}_k = D_\theta(s_k)$ and select the theory length

$$k^* = \underset{k \in \{1, \ldots, K+1\}}{\arg \min} \lambda_{\text{MDL}}^k \, \ell(y, \hat{y}_k), \qquad (5)$$

where $\lambda_{\text{MDL}} > 1$ controls the strength of the simplicity bias and penalizes longer programs exponentially. After selecting the explanation length $k^*$, the model is updated by backpropagating the loss in Eq. (3) only from the corresponding prediction $\hat{y}_{k^*}$.

### 3.4. Practical Implementation

**Deterministic execution.** In the general formulation, primitive execution is modeled as a stochastic state transition $p_\theta(s_{k+1} \mid s_k, z_{i_k})$. In this work, however, for simplicity and training stability, we adopt a deterministic special case and implement execution as $s_{k+1} = f_\theta(s_k, z_{i_k})$, which corresponds to a degenerate transition distribution with all probability mass concentrated at a single next state. Under deterministic execution, the likelihood reduces to a reconstruction loss $\ell(y, \hat{y}_\tau) \equiv -\log p_\theta(y \mid x, \tau)$ where $\hat{y}_\tau = D_\theta(f_{\tau,\theta}(E_\theta(x)))$.

**State Grounding.** A limitation of the formulation is that intermediate states $s_k$ for $k \leq K$ are not required to correspond to valid observations, as long as the final state $s_{K+1}$ reconstructs $y$. We find that this flexibility hinders the discovery of compositional program structure, since the model may learn shortcut transitions that are not useful as reusable building blocks. To address this issue, we ground each intermediate state to the encoder's latent space by enforcing consistency through a decode–encode cycle:

$$\mathcal{L}_{\text{state}} = \sum_{k=1}^{K} \left\| s_k - \text{sg}[E_\theta(D_\theta(s_k))] \right\|^2. \qquad (6)$$

Here, $\text{sg}[\cdot]$ denotes the stop-gradient operator. This loss updates only the transition model, encouraging each $s_k$ to lie on the manifold of valid latent representations.

**Training objective of the practical implementation.** When the discrete program $\tau$ is implemented using a VQ-VAE, the variational KL term in Eq. (3) is not computed explicitly; instead, discreteness is enforced through the standard codebook and commitment losses $\mathcal{L}_{\text{VQ}}$, resulting in the following objective:

$$\mathcal{L}_{\text{NEO}}(\theta, \phi) = \mathbb{E}_{q_{\phi,\theta}(\tau \mid x, y)}[\ell(y, \hat{y}_\tau)]$$
$$+ \lambda_{\text{vq}} \mathcal{L}_{\text{VQ}} + \lambda_{\text{state}} \mathcal{L}_{\text{state}}. \qquad (7)$$

Given the MDL-selected program length $k^*$ from Eq. (5), we optimize a truncated version of this objective by backpropagating only through the corresponding prediction $\hat{y}_{k^*}$ and ignoring all subsequent execution steps. This enforces the MDL principle during learning by encouraging the

model to explain each phenomenon using the shortest accurate program. Additionally, we use pretrained parameters for the encoder and decoder for simplicity and stability. A detailed description of the pretrained model is in Appendix F.

### 3.5. Inference at Test Time

At test time, NEO infers a theory for a previously unseen phenomenon $(x, y)$ by iteratively constructing a program through the theory programmer. Starting from the initial latent state $s_1 = E_\theta(x)$, the model sequentially selects primitive operations according to $q_\phi(z_{i_k} \mid s_k, y)$ and applies the transition model to obtain $s_{k+1} = f_\theta(s_k, z_{i_k})$. Execution is terminated when the reconstruction error falls below a predefined threshold, $\ell(y, \hat{y}_k) \leq \varepsilon$, at which point the current program is returned as the inferred theory.

Because programs are constructed compositionally, this procedure naturally supports *length generalization*: the model can generate programs longer than those observed during training simply by continuing the same primitive composition process. Moreover, inference permits explicit *intervention* by overriding the theory programmer's primitive selection, enabling exploration of counterfactual execution traces and the discovery of novel program trajectories. Pseudocodes for both training and inference are provided in Appendix B.

## 4. Related Works

In this section, we briefly discuss the related works while providing a more detail discussion in the Appendix G. Our work is related to several lines of research. First, latent action and world models aim to learn compact latent dynamics from observation-only data, including latent action models such as LAPO (Schmidt & Jiang, 2024), AdaWorld (Gao et al., 2025), LAPA (Ye et al., 2025), and Genie (Bruce et al., 2024), as well as world models such as RSSM, and Dreamer (Hafner et al., 2018; 2019; 2020; 2024). Second, abstract reasoning benchmarks such as ARC-AGI (Chollet, 2019; Chollet et al., 2025; 2026) study program induction from few demonstrations under fixed representational biases. Third, neural program induction and synthesis approaches, including NTM (Graves et al., 2014), NPI (Reed & de Freitas, 2016), LEAPS (Trivedi et al., 2022), HPRL (Liu et al., 2023), LPN (Macfarlane & Bonnet, 2025), and Dream-Coder (Ellis et al., 2020), investigate learning executable programs from input–output behavior, typically within predefined symbolic program spaces. Finally, compositional representation learning has been explored through adaptive tokenization (Duggal et al., 2024; 2025) and emergent communication methods (Elberg et al., 2025), which learn variable-length, compositional descriptions.

Table 1. **Performance comparison on the GridWorld environment.** Results show mean across three runs for each metric. NEO-S is evaluated with $B = 64$ budget.

| $\alpha$ | Method | In-distribution | | Comp. OOD | | Length OOD | |
|---|---|---|---|---|---|---|---|
| | | Self-Ex. | Transf. | Self-Ex. | Transf. | Self-Ex. | Transf. |
| 0.33 | Disc-Mono | 0.988 | **0.983** | 0.000 | 0.000 | 0.000 | 0.000 |
| | Cont-Mono | 0.975 | 0.001 | 0.431 | 0.000 | 0.053 | 0.000 |
| | Cont-Mono-Opt | **0.994** | 0.000 | 0.726 | 0.000 | 0.209 | 0.000 |
| | **NEO (Ours)** | 0.914 | 0.911 | 0.934 | 0.933 | 0.853 | 0.845 |
| | **NEO-S (Ours)** | 0.993 | 0.970 | **0.995** | **0.976** | **0.978** | **0.907** |
| 0.66 | Disc-Mono | 0.978 | 0.973 | 0.000 | 0.000 | 0.000 | 0.000 |
| | Cont-Mono | 0.979 | 0.000 | 0.929 | 0.000 | 0.531 | 0.000 |
| | Cont-Mono-Opt | 0.991 | 0.000 | 0.972 | 0.000 | 0.805 | 0.001 |
| | **NEO (Ours)** | 0.966 | 0.965 | 0.964 | 0.963 | 0.930 | 0.927 |
| | **NEO-S (Ours)** | **0.997** | **0.987** | **0.998** | **0.987** | **0.991** | **0.949** |
| 1.00 | Disc-Mono | 0.928 | 0.921 | . | . | 0.000 | 0.000 |
| | Cont-Mono | 0.982 | 0.000 | . | . | 0.673 | 0.000 |
| | Cont-Mono-Opt | 0.992 | 0.000 | . | . | 0.877 | 0.001 |
| | **NEO (Ours)** | 0.953 | 0.949 | . | . | 0.902 | 0.898 |
| | **NEO-S (Ours)** | **0.995** | **0.975** | . | . | **0.986** | **0.926** |

Table 2. **Performance comparison on the Arithmetic Factorization Reasoning task.** Results show mean across three runs for each metric. NEO-S is evaluated with $B = 1024$ budget.

| $\alpha$ | Method | In-distribution | | Comp. OOD | | Length OOD | |
|---|---|---|---|---|---|---|---|
| | | Self-Ex. | Transf. | Self-Ex. | Transf. | Self-Ex. | Transf. |
| 0.33 | Disc-Mono | **0.890** | 0.668 | 0.005 | 0.004 | 0.012 | 0.009 |
| | Cont-Mono | 0.834 | 0.001 | 0.165 | 0.000 | 0.321 | 0.000 |
| | Cont-Mono-Opt | 0.866 | 0.001 | 0.218 | 0.000 | 0.394 | 0.000 |
| | **NEO (Ours)** | 0.847 | 0.792 | 0.357 | 0.345 | 0.045 | 0.038 |
| | **NEO-S (Ours)** | 0.857 | **0.809** | **0.831** | **0.759** | **0.620** | **0.524** |
| 0.66 | Disc-Mono | 0.737 | 0.572 | 0.005 | 0.002 | 0.006 | 0.004 |
| | Cont-Mono | 0.500 | 0.002 | 0.086 | 0.000 | 0.158 | 0.001 |
| | Cont-Mono-Opt | 0.565 | 0.002 | 0.122 | 0.000 | 0.216 | 0.001 |
| | **NEO (Ours)** | 0.794 | 0.731 | 0.609 | 0.573 | 0.023 | 0.019 |
| | **NEO-S (Ours)** | **0.977** | **0.939** | **0.994** | **0.959** | **0.766** | **0.696** |
| 1.00 | Disc-Mono | 0.662 | 0.475 | . | . | 0.006 | 0.004 |
| | Cont-Mono | 0.810 | 0.000 | . | . | 0.649 | 0.000 |
| | Cont-Mono-Opt | 0.846 | 0.000 | . | . | 0.743 | 0.000 |
| | **NEO (Ours)** | 0.724 | 0.675 | . | . | 0.025 | 0.023 |
| | **NEO-S (Ours)** | **0.990** | **0.954** | . | . | **0.799** | **0.707** |

# 5. Experiments

Our experiments evaluate whether NEO can (1) discover latent primitive operations that are never directly observed during training and (2) explain dynamics arising from previously unseen program compositions. To this end, we introduce the Observation-to-Theory Induction Benchmark.

## 5.1. Observation to Theory Induction Benchmark

**O**bservation-to-**T**heory **I**nduction **B**enchmark (OTIB) evaluates whether a model can infer reusable primitives from raw observation pairs $(x, y)$ without supervision. Its central criterion is *transferable explanation*: a theory induced from one transition should generalize to new inputs, rather than memorizing instance-specific mappings.

We define a training set and three evaluation sets: In-Distribution (ID) test, compositional Out-of-Distribution (OOD), and length OOD. Following SVIB-style (Kim et al., 2023) compositional generalization, we consider all program compositions within an observable complexity range and include an $\alpha$ fraction of them in the training set (ID); the remaining compositions in this range form the compositional OOD set. We use $\alpha \in \{0.33, 0.66, 1.00\}$ throughout our experiments. Our sampling ensures that some primitives are never observed in isolation and instead appear only as parts of longer, entangled programs, requiring models to discover them by decomposing multi-step hidden transitions; smaller $\alpha$ increases this difficulty. By construction, the training compositions alone contain sufficient evidence, in principle, to recover the full primitive set via such decomposition.

Each evaluation instance consists of a *support* pair $(x^{(1)}, y^{(1)})$ and a *query* pair $(x^{(2)}, y^{(2)})$ generated by the same latent program $\tau$. Given the support pair, a model induces a theory $\hat{\tau}$ from $(x^{(1)}, y^{(1)})$. The induced theory is

then executed on $x^{(1)}$ and $x^{(2)}$ to produce predictions $\hat{y}^{(1)}$ and $\hat{y}^{(2)}$. We define *self-explainability* as $d(\hat{y}^{(1)}, y^{(1)})$ and *transferability* as $d(\hat{y}^{(2)}, y^{(2)})$, where $d$ is a domain-specific evaluation metric on outputs. Transferability specifically tests whether the induced theory is reusable (i.e., generalizes to new inputs) rather than encoding instance-specific information about $y^{(1)}$.

We instantiate OTIB in three domains: GridWorld, Arithmetic Reasoning, and Image Editing.

**GridWorld** is a controlled $10 \times 10$ environment where an object moves via latent motion primitives (up/down/left/right). Training uses programs of length 1–3. We report compositional OOD transfer under the resulting $\alpha$-splits and evaluate extrapolation to longer programs up to length 8. See Appendix C.3 for details.

**Arithmetic Factorization Reasoning** is a symbolic reasoning benchmark. Each observation is an integer pair $(x, y)$, where $y$ is produced by applying a sequence of primitives from $\{\times 2, \times 3, \times 5, \times 7\}$ to $x$. We train on length 1–3 programs and use $\alpha$ to vary which compositions are included in training versus held out. Evaluation covers compositional OOD within lengths 1–3 and length OOD generalization to programs of length 4–6. Details appear in Appendix C.5.

**Image Editing** is a visual transformation task on CIFAR-10, where primitives correspond to 8 editing operations (e.g., rotation, brightness adjustment, masking). Models are trained on compositions of length 1–2. We then test compositional OOD transfer within lengths 1–2 and generalization to longer edits of length 3–4. Implementation details are deferred to Appendix C.6.

## 5.2. Baselines

**Discrete Monolithic (Disc-Mono).** A conditional VQ-VAE that auto-encodes $y$ conditioned on $x$, representing the pro-

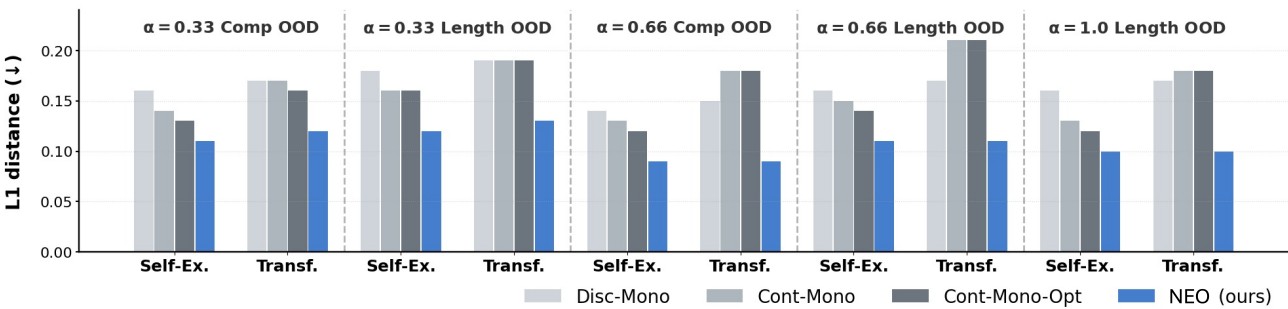

*Figure 3.* **Comparison of image-editing performance across $\alpha$-controlled dataset complexity and OOD settings, including length OOD.** NEO consistently outperforms baselines across all $\alpha$-controlled OOD regimes and length OOD, for both self-explainability and transferability, as measured by the $\ell_1$ distance between the predicted image $\hat{y}$ and the ground-truth target $y$ (lower is better).

gram as a single quantized vector. This corresponds to latent action models such as LAPO (Schmidt & Jiang, 2024) and Genie (Bruce et al., 2024).

**Continuous Monolithic (Cont-Mono).** A conditional $\beta$-VAE (Higgins et al., 2016) that represents the program as a single continuous latent vector $z \in \mathbb{R}^d$. This corresponds to a latent action model such as AdaWorld (Gao et al., 2025).

**Continuous Monolithic with Program Optimization (Cont-Mono-Opt).** This extends Cont-Mono by iteratively refining the program vector $z$ via gradient ascent on $\log p(y \mid x, z)$, providing a stronger baseline with test-time search, inspired by LPN (Macfarlane & Bonnet, 2025).

Note that while some baselines correspond to latent action models, our focus is not on learning actions for control but on discovering more abstract compositional primitives for explanation and theory construction. We provide further experimental details in Appendix C, D.3.

### 5.3. GridWorld Results

Table 1 reports performance across $\alpha$-ratings on in-distribution, compositional OOD, and length OOD tasks. NEO consistently outperforms baselines, which largely fail to generalize. Disc-Mono transfers well in-distribution (e.g., 0.983 at $\alpha = 0.33$) but collapses on OOD, suggesting that single vector programs do not yield compositional understanding. Cont-Mono scores high on self-explainability yet still show near-zero transfer, consistent with encoding y-specific information in the latent rather than inducing a reusable theory. In contrast, NEO maintains strong OOD transfer even at the hardest setting (e.g., 0.933, 0.845 on compositional, length OOD at $\alpha = 0.33$), highlighting the importance of Learning to Theorize. To further examine the test-time scalability of our approach, we introduce NEO-S, which augments NEO with a sampling-based test-time search procedure. NEO-S further improves transferability with test-time search; see Sec. 5.6 and Appendix D.1 for details and additional comparative results.

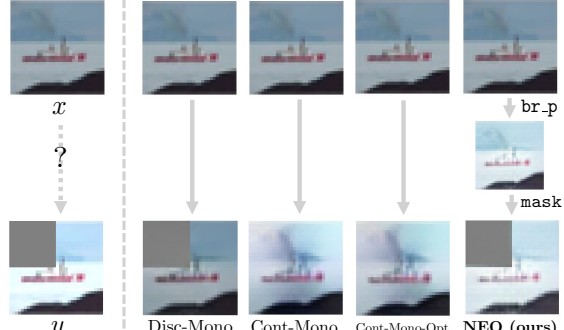

*Figure 4.* **Visualization of explanations for a compositional OOD in the image-editing task ($\alpha = 0.66$).** The leftmost column shows the observed source–target pair $(x, y)$. Baseline models generate $y$ via a single-step prediction or by relying on action combinations observed only in the in-distribution data, and thus fail to decompose the novel OOD transformation. In contrast, NEO explains the same phenomenon as a sequence of learned primitive actions, enabling systematic OOD generalization through explicit compositional explanations.

### 5.4. Arithmetic Factorization Reasoning Results

Table 2 reports performance on Arithmetic Reasoning across different $\alpha$-ratings. NEO consistently outperforms baseline methods, demonstrating strong compositional generalization even under partial training coverage (e.g., 0.345 at $\alpha$=0.33 and 0.573 at $\alpha$=0.66), indicating that it successfully acquires reusable multiplicative primitives (see Sec. 5.6 for more details).

Note that on this task, length OOD generalization is substantially more challenging: transitions such as $x$=73 $\rightarrow$ $y$=273, 750 require inferring a six-step factorization ($\times 5 \times 5 \times 5 \times 5 \times 3 \times 2$) and executing exact multi-digit arithmetic. Consequently, when relying solely on the learned policy, NEO attains low length-OOD transfer accuracy (0.019–0.038) but still outperforms monolithic baselines (near-zero). Importantly, this limitation does not stem from missing primitives but from the difficulty of selecting and composing them correctly over longer horizons. The search-based counterpart, NEO-S, addresses this by performing

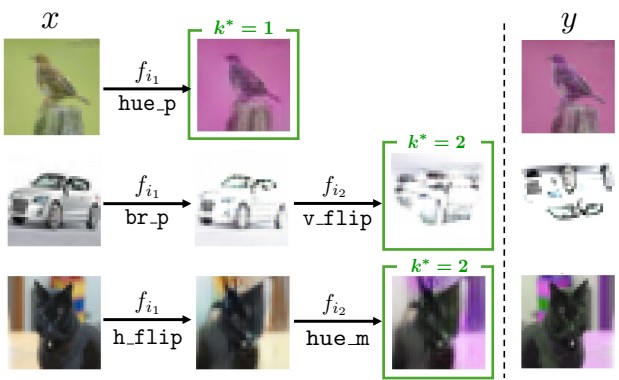

*Figure 5.* **Visualization of instance-wise program length selection under the MDL principle.** For each instance, the model selects an optimal program length $k^*$ that aligns with the ground-truth number of underlying transitions, demonstrating adaptive explanation length rather than a fixed horizon. In addition, the selected programs recover semantically correct action sequences; see Sec. C.6.1 for details on primitive definitions.

test-time search over program compositions (Fig. 6(b)), leading to dramatic performance gains. Test-time scaling boosts length-OOD accuracy from approximately 0.02 to 0.696 at $\alpha$=0.66 and 0.707 at $\alpha$=1.00. The large gains from test-time scaling suggest that NEO already learns the required primitives, and additional search mainly improves their long-horizon recomposition.

### 5.5. Image Editing Results

Figure 3 shows that NEO consistently achieves the lowest L1 distance ($\downarrow$) on both compositional OOD and length OOD across all $\alpha$-ratings, indicating robust programmatic understanding in high-dimensional continuous pixel space. Figure 4 illustrates how NEO explains compositional ood program: while baselines generate $y$ via a single monolithic vector program, NEO composes reusable primitives into an executable theory (e.g., *brightness*+ followed by *mask*), yielding faithful explanations even for previously unseen phenomena. Finally, Figure 5 shows that NEO can automatically select an optimal explanation length $k^*$, adapting the complexity of the induced program to that of the underlying transformation (see Appendix C.7 for more results.).

### 5.6. Analysis

**Discovering Unseen Primitives.** Figure 7 reports the *primitiveness* of learned codes, measured by how well the primitives induced by a model align with the ground-truth (GT) primitive set (See Appendix C.2.2 for a detailed definition). The GT bar reflects the fraction of primitives that are directly observable in training observations (relative to the full primitive set); thus, a model that simply memorizes what is observable can only appear comparable to GT. In contrast, NEO consistently achieves much higher score—often approaching the full primitive set—even when only a small

*Table 3.* **Ablations on grounding loss, codebook size ($|\mathcal{E}|$), and** $\lambda_{\text{MDL}}$. Without grounding loss, intermediate programs drift from meaningful state space, leading to degradation. The codebook size and the MDL weight control the expressive capacity of the program space and the pressure toward shorter explanations, respectively, inducing a trade-off between expressivity and program length.

| Method | Prim. | In-distribution | | Comp. OOD | | Length OOD | |
|---|---|---|---|---|---|---|---|
| | | Self-Ex. | Transf. | Self-Ex. | Transf. | Self-Ex. | Transf. |
| **NEO (Base)** | **1.000** | 0.914 | 0.911 | 0.934 | 0.933 | 0.853 | 0.845 |
| - No Grounding | 0.002 | 0.000 | 0.000 | 0.000 | 0.000 | 0.000 | 0.000 |
| **NEO + $|\mathcal{E}|$=36** | **1.000** | **0.962** | **0.956** | **0.980** | **0.976** | **0.935** | **0.930** |
| - w/ $\lambda_{\text{MDL}}$=1.2 | 0.213 | 0.732 | 0.621 | 0.228 | 0.160 | 0.221 | 0.169 |
| - w/ $\lambda_{\text{MDL}}$=0.8 | 1.000 | 0.916 | 0.856 | 0.859 | 0.956 | 0.750 | 0.748 |

subset is directly observed (low $\alpha$). This suggests that NEO resolves theories into finer, reusable primitive units, providing the appropriate compositional building blocks.

**Scaling at Test-time.** Since the theory programmer $q_\phi(z \mid s, y)$ is probabilistic, it can generate multiple plausible theories from a single observation $x \to y$, enabling test-time search (denoted as NEO-S). Given a sampling budget $B$, we draw $B$ candidate theories and select a single theory via majority voting (i.e., the most consistently supported candidate among the samples for explaining the same $x \to y$ pair). We then measure transferability by evaluating whether this one selected theory generalizes to new inputs beyond the original observation. Figure 6(a) shows that increasing $B$ consistently improves both self-explainability and transferability, while monolithic baselines remain flat—demonstrating that compositional structure enables effective test-time search.

Furthermore, because learned primitives are reusable and composable, the program executor enables intervention at the primitive level. By increasing the sampling temperature, we choose the primitive to intervene the world model, enabling exploratory compositions—simulating transitions only through latent states never encountered during training. Figure 6 (b) visualizes how sampling-based search expands the theory programmer's choices on Arithmetic Factorization: the blue edges trace the default argmax selections made by the theory programmer, while the orange dashed edges are alternative primitives sampled via exploration. This expands the search over program compositions from the same $x \to y$ pair and increases the chance of finding a correct execution path within a limited search budget. Figure 20 shows that this exploration further improves performance when combined with sufficient budget.

### 5.7. Ablation Studies

To better understand what drives NEO's strong empirical behavior, we analyze key design choices: state grounding loss and codebook size $|\mathcal{E}|$ with MDL weight $\lambda_{\text{MDL}}$.

**State grounding is essential.** We found that removing state grounding causes training to collapse, with primitive-

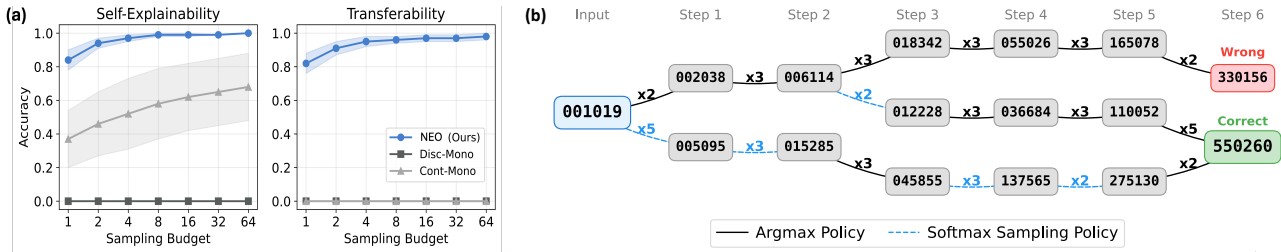

*Figure 6.* (a) **Test-time scaling via sampling on GridWorld**. As the sampling budget increases, NEO approaches near-perfect accuracy, while monolithic baselines fail to improve. Shaded regions show variability across runs. (b) **Execution paths of sampled programs on the Arithmetic Factorization Reasoning task**. Test-time scaling is achieved by sampling diverse compositions of reusable learned primitives. Black solid lines denote argmax selections by theory programmer; blue dashed lines denote sampled selctions from softmax distribution induced by theory programmer. See Appendix E for more examples.

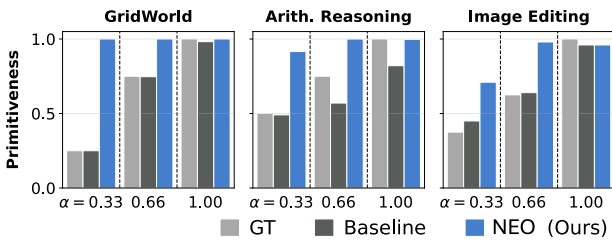

*Figure 7.* **Primitiveness of learned codebook across tasks and dataset complexity ($\alpha$).** GT denotes the maximum achievable primitiveness only with directly observed primitves.

ness dropping to $0.002$ and both self-explainability and transferability becoming zero across all splits. This suggests that grounding anchors each intermediate state back to the model's state manifold, ensuring that subsequent primitive operations are applied within a consistent representation rather than drifting off-manifold into unstable latents.

**Resolution of Theories.** We also examine codebook size $|\mathcal{E}|$ and the MDL weight $\lambda_{\mathrm{MDL}}$. To rule out concerns that our results depend on choosing $|\mathcal{E}|$ close to the ground truth, we additionally test a highly over-complete codebook. Even with substantial over-capacity (e.g., $|\mathcal{E}| = 36$), where the model could in principle allocate separate codes to observed training compositions, NEO instead induces primitive-level codes and composes them into multi-step explanations. However, $\lambda_{\mathrm{MDL}}$ crucially shapes the learning dynamics: when $\lambda_{\mathrm{MDL}}$ is too large (e.g., $1.2$), the model is incentivized to adopt overly short, entangled program explanations, yielding low primitiveness and poor transferability. Figure 8 clarifies this effect: the dashed GT line denotes the program length obtained when each training composition is explained using the ground-truth primitive program, and $\lambda_{\mathrm{MDL}} \in \{0.8, 1.0\}$ yields explanation lengths that closely track this GT length. Together with primitiveness=1.0 in Table 3, this indicates that NEO recovers the full set of underlying primitives—including those never directly observed—and that the theory programmer composes them into multi-step programs rather than memorizing observed

compositions. Additional details, including visualizations of the role of each code, are provided in Appendix C.4.

## 6. Limitations & Discussion

This work should be viewed as an initial proof of concept for *Learning-to-Theorize*. The current formulation assumes a relatively small, discrete set of primitives and short program lengths, which limits its scalability to domains with long-horizon, continuous, or highly structured dynamics. Moreover, primitive semantics are induced only through reconstruction, and therefore are not guaranteed to align with human-interpretable concepts or truly causal factors. Our inference procedure also relies on deterministic execution and reconstruction-based stopping criteria, which may be brittle under noise, ambiguity, or partial observability. Finally, our experiments are restricted to controlled synthetic benchmarks. Extending L2T to richer, real-world environments with complex perceptual inputs, stochastic dynamics, and open-ended theory spaces remains an important direction for future work.

## 7. Conclusion

We introduced *Learning-to-Theorize*, a learning paradigm in which models acquire explanatory theories by inducing executable, compositional programs from observation. We instantiated this paradigm with the *Neural Theorizer (NEO)*, a probabilistic neural model that learns reusable primitives and composes them into latent programs to explain observed phenomena. By representing theories as programs and regulating their complexity through the MDL principle, NEO achieves explanation-driven generalization to unseen program compositions and to programs longer than those observed during training. Although our results are limited to controlled settings, they provide a proof of concept that structured, programmatic theories can be learned from raw observations. More broadly, our findings point toward world models that move beyond prediction-centric learning toward explanatory, compositional understanding.

## Acknowledgements

This research was supported by the Brain Pool Plus Program (No. 2021H1D3A2A03103645) and the GRDC (Global Research Development Center) Cooperative Hub Program (RS-202400436165) through the National Research Foundation of Korea (NRF), funded by the Ministry of Science and ICT (MSIT). The authors thank all the members of the Machine Learning and Mind Lab (MLML) for their helpful discussions and support. In particular, we are grateful to Hyeonseo Cho, Minsu Kim, Mingyu Jo, Seungju Back, and Junyeong Park for their valuable feedback and encouragement. SJ thanks Yoshua Bengio for helpful discussions.

## Impact Statement

This work introduces a new learning paradigm, Learning-to-Theorize, and a model that induces executable theories from observation. By shifting learning from direct prediction toward the discovery of reusable explanatory structure, this research may contribute to improved generalization, interpretability, and abstraction in future AI systems, with potential relevance to scientific modeling and world modeling. The work is primarily methodological and does not target applications involving human subjects, personal data, or automated decision-making. As with other representation-learning methods, the proposed approach could be misused if applied without appropriate safeguards; however, we do not identify risks specific to this contribution beyond those generally associated with machine learning research. We emphasize that this work is intended as a foundational study of learning mechanisms rather than a deployment-oriented system.

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

## A. Use of Large Language Models

During the preparation of this manuscript, the authors used large language models (e.g., Claude, GPT-4) to assist with refining prose, improving grammatical clarity, and enhancing readability. These tools were not used for generating scientific ideas, experimental design, data analysis, or drawing conclusions. All content was critically reviewed, verified, and revised by the authors, who take full responsibility for the final manuscript.

## B. Pseudocode of NEO

---

**Algorithm 1** Training NEO (Neural Theorizer)

---

**Require:** Training set $\mathcal{D}_{\text{train}} = \{(x_n, y_n)\}_{n=1}^N$, program prior $p(\tau)$, number of unroll steps $K$, learning rate $\eta$
**Require:** Encoder $E_\theta$, decoder $D_\theta$, program inference network $q_\phi(\tau \mid x, y)$, program execution network $f_\theta(s, z)$
1: **for** each training iteration **do**
2:     Sample a minibatch $\{(x, y)\}$ from $\mathcal{D}_{\text{train}}$
3:     Encode observations: $s_1 \leftarrow E_\theta(x)$, $s_y \leftarrow E_\theta(y)$
4:     **for** $k = 1$ **to** $K$ **do**
5:         Sample a primitive: $z_{i_k} \sim q_\phi(z_{i_k} \mid s_k, s_y)$
6:         Execute primitive: $s_{k+1} \leftarrow f_\theta(s_k, z_{i_k})$
7:     **end for**
8:     Length selection: $k^* \leftarrow \arg\min_k \lambda_{\text{MDL}}^k \cdot \ell(y, D_\theta(s_k))$
9:     Reconstruction loss: $\mathcal{L}_{\text{rec}} \leftarrow \ell(y, D_\theta(s_{k^*}))$
10:    State grounding loss: $\mathcal{L}_{\text{state}} = \sum_{k=1}^K \|s_k - \text{sg}[E_\theta(D_\theta(s_k))]\|^2$
11:    Total loss: $\mathcal{L} \leftarrow \mathcal{L}_{\text{rec}} + \lambda_{\text{vq}}\mathcal{L}_{\text{vq}} + \lambda_{\text{state}}\mathcal{L}_{\text{state}}$
12:    Update parameters: $(\phi, \theta) \leftarrow (\phi, \theta) - \eta\nabla_{\phi,\theta}\mathcal{L}$
13: **end for**

---

---

**Algorithm 2** Inference with NEO: Transfer Evaluation

---

**Require:** Test example $(x_s, y_s, x_q, y_q)$ (support input/output, query input/target)
**Require:** Trained encoder $E_\theta$, decoder $D_\theta$, program inference $q_\phi$, execution $f_\theta$
**Require:** Max program unroll steps $K$, MDL coefficient $\lambda_{\text{MDL}}$
1: **// Phase 1: Support rollout (program extraction)**
2: $s_1 \leftarrow E_\theta(x_s)$,    $s_y \leftarrow E_\theta(y_s)$
3: **for** $k = 1$ **to** $K$ **do**
4:    $z_{i_k} \leftarrow \arg\max_z \ q_\phi(z \mid s_k, s_y)$ {greedy action selection}
5:    $s_{k+1} \leftarrow f_\theta(s_k, z_{i_k})$
6: **end for**
7: **// Length selection (shortest correct / MDL):**
8: $k^* \leftarrow \arg\min_k \ \lambda_{\text{MDL}}^k \cdot \ell(y_s, D_\theta(s_{k+1}))$ {prefer shorter correct programs}
9: Extracted program: $\tau^* = (z_{i_1}, \ldots, z_{i_{k^*}})$
10:
11: **// Phase 2: Transfer rollout (program application to query)**
12: $s_1' \leftarrow E_\theta(x_q)$
13: **for** $k = 1$ **to** $k^*$ **do**
14:    $s_{k+1}' \leftarrow f_\theta(s_k', z_{i_k})$ {reuse extracted primitives}
15: **end for**
16: $\hat{y}_q \leftarrow D_\theta(s_{k^*+1}')$
17: **return** $\hat{y}_q$,    correct $\leftarrow (\hat{y}_q = y_q)$

---

---

**Algorithm 3** Test-Time Scaling with NEO (select@$B$)

---

**Require:** Test example $(x_s, y_s, x_q, y_q)$, sampling budget $B$, temperature $T$
**Require:** Trained encoder $E_\theta$, decoder $D_\theta$, program inference network $q_\phi$, execution network $f_\theta$, codebook $\mathcal{C}$
**Require:** Max unroll steps $K$
 1: **// Phase 1: Sample $B$ programs on support pair**
 2: $s_1 \leftarrow E_\theta(x_s), \quad s_y \leftarrow E_\theta(y_s)$
 3: **for** $b = 1$ **to** $B$ **do**
 4: $\quad s_1^{(b)} \leftarrow s_1$
 5: $\quad$ **for** $k = 1$ **to** $K$ **do**
 6: $\quad\quad$ Compute logits: $\ell_j = -\|q_\phi(s_k^{(b)}, s_y) - c_j\|^2 / T$ for each $c_j \in \mathcal{C}$
 7: $\quad\quad$ Sample: $z_{i_k}^{(b)} \sim \text{Softmax}(\ell)$ {temperature-scaled sampling}
 8: $\quad\quad s_{k+1}^{(b)} \leftarrow f_\theta(s_k^{(b)}, z_{i_k}^{(b)})$
 9: $\quad$ **end for**
10: $\quad$ Length selection: $k_b^* \leftarrow \arg\min_k \ \lambda_{\text{MDL}}^k \cdot \ell(y_s, D_\theta(s_{k+1}^{(b)}))$
11: $\quad \tau^{(b)} \leftarrow (z_{i_1}^{(b)}, \ldots, z_{i_{k_b^*}}^{(b)})$
12: **end for**
13:
14: **// Phase 2: Filter and select**
15: $\mathcal{S} \leftarrow \{b \mid D_\theta(s_{k_b^*+1}^{(b)}) = y_s\}$
16: **if** $\mathcal{S} \neq \emptyset$ **then**
17: $\quad b^* \leftarrow \arg\max_{b \in \mathcal{S}} |\{b' \in \mathcal{S} \mid \tau^{(b')} = \tau^{(b)}\}|$ {most frequent program}
18: **else**
19: $\quad$ **return** failure
20: **end if**
21:
22: **// Phase 3: Transfer**
23: $s_1' \leftarrow E_\theta(x_q)$
24: **for** $k = 1$ **to** $k_{b^*}^*$ **do**
25: $\quad s_{k+1}' \leftarrow f_\theta(s_k', z_{i_k}^{(b^*)})$
26: **end for**
27: $\hat{y}_q \leftarrow D_\theta(s_{k_{b^*}^*+1}')$
28: **return** $\hat{y}_q, \quad$ correct $\leftarrow (\hat{y}_q = y_q)$

---

# C. Additional Experimental Details

## C.1. Baselines

To evaluate the benefit of compositional program structure, we compare NEO against three baselines that represent programs as monolithic vectors rather than as sequences of primitives. These baselines span different design choices: whether the program representation is discrete or continuous, and whether inference is amortized or optimized at test time. Importantly, while some baselines correspond to architectures used in latent action models for control, our focus is not on learning actions for policy execution but on discovering compositional primitives for explanation and theory construction.

**Discrete Monolithic (Disc-Mono).** This baseline represents a program as a single quantized vector using a conditional VQ-VAE architecture (Van Den Oord et al., 2017). Given an observation pair $(x, y)$, the encoder maps the transformation into a discrete codebook entry $z \in \mathcal{E} = \{e_1, e_2, \ldots, e_{|\mathcal{E}|}\}$, where $|\mathcal{E}|$ is the codebook size. The program is thus a single discrete symbol selected from a finite vocabulary, with no internal compositional structure. The decoder reconstructs $y$ from $x$ and the selected codebook entry $z$. Training follows the standard VQ-VAE objective with commitment loss and codebook update via codebook loss or exponential moving average. This architecture corresponds to latent action models such as LAPO (Schmidt & Jiang, 2024) and Genie (Bruce et al., 2024), which learn discrete action representations from observation pairs. This baseline tests whether a discrete but non-compositional representation suffices for generalization to unseen program compositions.

**Continuous Monolithic (Cont-Mono).** This baseline represents a program as a single continuous latent vector using a conditional $\beta$-VAE architecture (Higgins et al., 2016). Given an observation pair $(x, y)$, the encoder produces a Gaussian posterior $q_\phi(z|x,y) = \mathcal{N}(\mu_\phi(x,y), \sigma_\phi^2(x,y))$, where $z \in \mathbb{R}^d$ serves as the program representation. The decoder reconstructs $y$ from $x$ and $z$. Training maximizes the ELBO:

$$\mathcal{L} = \mathbb{E}_{q_\phi(z|x,y)}[\log p_\theta(y|x,z)] - \beta \operatorname{KL}(q_\phi(z|x,y)\|p(z)), \tag{8}$$

where $p(z) = \mathcal{N}(0, I)$ is a standard Gaussian prior and $\beta$ controls the strength of disentanglement pressure. Inference is fully amortized: a single forward pass through the encoder produces $z$ without iterative refinement. This architecture corresponds to world models such as AdaWorld (Gao et al., 2025), which learn continuous latent dynamics from observations. This baseline tests whether continuous representations without compositional structure can capture transferable transformations.

**Continuous Monolithic with Optimization (Cont-Mono-Opt).** This baseline extends Cont-Mono by optimizing the program vector at test time rather than relying solely on amortized inference. Given a phenomenon $(x, y)$, the latent vector $z$ is first initialized from the amortized encoder and then refined via gradient ascent on the reconstruction objective:

$$z^{(t+1)} = z^{(t)} + \eta \nabla_z \log p_\theta(y|x, z^{(t)}), \tag{9}$$

where $\eta$ is the learning rate and the optimization runs for a fixed number of steps. This provides a stronger baseline by allowing the model to search for a better explanation at test time, while still lacking compositional structure. The architecture corresponds to approaches such as Latent Program Network (LPN) (Macfarlane & Bonnet, 2025), which employ latent optimization for improved inference. This baseline tests whether test-time optimization over a continuous latent space can compensate for the absence of primitive decomposition, and whether the limitation of monolithic baselines stems from amortization gap or from the lack of compositional structure itself.

### C.2. Evaluation Metrics

#### C.2.1. CODE–PRIMITIVE ALIGNMENT METRIC

In this section, we report the code–primitive alignment metric, which quantifies how well each learned code in the codebook corresponds to a ground-truth primitive operation. The goal of this metric is to evaluate whether the learned codes capture meaningful and reusable primitives, rather than entangled or ambiguous transformations.

Formally, let $\mathcal{Z} = z_i$ denote the set of learned codes and $\mathcal{A} = a_j$ the set of ground-truth primitives. Given a test set of $N$ inputs $(x_n)_{n=1}^N$, we measure the alignment between code $z_i$ and primitive $a_j$ by counting how often applying $z_i$ to $x_n$ produces an outcome $y_n^{a_j}$ consistent with the effect of $a_j$:

$$C_{i,j} = \sum_{n=1}^N \mathbf{1}\big[\mathcal{M}(z_i, x_n, y_n^{a_j}, a_j)\big],$$

where $\mathcal{M}(\cdot)$ is a task-dependent matching criterion. The resulting matrix $C \in \mathbb{R}^{|\mathcal{Z}| \times |\mathcal{A}|}$ forms the code–primitive alignment matrix.

The matching criterion differs across domains. (1) In GridWorld and Arithmetic Reasoning, where exact correctness is well-defined, $\mathcal{M}$ counts a match if the predicted output exactly matches the ground-truth output at all pixels (or symbols). (2) In Image Editing, where exact matches are ambiguous, we compare the output generated by each code against all candidate ground-truth primitive combinations and count the primitive that yields the minimum reconstruction loss, resulting in a soft alignment based on loss minimization.

#### C.2.2. ACTION PRIMITIVENESS

To measure how completely the learned action space can reproduce the atomic (primitive) transformations of the environment, we introduce the *Action Primitiveness* metric.

**Primitive Evaluation Dataset.** We construct an evaluation dataset $\mathcal{D}_{\text{prim}}$ where each ground-truth primitive action $g \in \mathcal{G}$ is applied exactly once:

$$\mathcal{D}_{\text{prim}} = \{(x, y) \mid y = \operatorname{Oracle}(x, g),\ g \in \mathcal{G}\} \tag{10}$$

This dataset serves as a comprehensive testbed to verify whether the model can replicate every atomic transformation in the environment.

**Primitiveness Computation.**   For each input-output pair $(x, y) \in \mathcal{D}_{\text{prim}}$, we check whether there exists at least one learned action $a \in \mathcal{A}$ that can produce an output identical to the target $y$:

$$\text{Primitiveness} = \frac{1}{|\mathcal{D}_{\text{prim}}|} \sum_{(x,y) \in \mathcal{D}_{\text{prim}}} \max_{a \in \mathcal{A}} \mathbf{1}[\hat{y}_a = y] \tag{11}$$

where $\hat{y}_a = \text{Dec}(p_\psi(\text{Enc}(x), a))$ is the reconstructed output obtained by applying learned action $a$ through the transition model, and $\mathbf{1}[\cdot]$ is the indicator function that returns 1 if the condition is satisfied and 0 otherwise. The equality $\hat{y}_a = y$ requires that all cells (pixels) of the generated output exactly match the target.

**Interpretation.**   Action Primitiveness measures the *coverage* of the learned action space over the ground-truth primitive actions. A score of 1.0 indicates that the model has learned at least one action capable of reproducing each oracle primitive, ensuring complete expressiveness. Conversely, a lower score reveals gaps in the learned action repertoire, where certain atomic transformations cannot be replicated by any single learned action.

This metric complements Action Purity: while Purity measures whether each learned action corresponds to a *single* oracle action (one-to-one correspondence), Primitiveness measures whether *all* oracle actions are covered by the learned action space (completeness).

### C.3. GridWorld Details

#### C.3.1. GRIDWORLD PRIMITIVES

In GridWorld, the environment consists of a $10 \times 10$ grid with a single object. The object can be acted on by four ground-truth motion primitives as shown in Table 4

*Table 4.* Ground-truth primitives used in the GridWorld task.

| Primitive | Description |
|---|---|
| move_up | Move the object one cell upward |
| move_right | Move the object one cell to the right |
| move_down | Move the object one cell downward |
| move_left | Move the object one cell to the left |

#### C.3.2. GRIDWORLD TEST SETTING

**Dataset Construction.**   We construct OTIB-GridWorld splits over *short* programs of length 1–3 ground-truth primitive steps, and evaluate **length OOD** using *longer* programs of length 4–8 steps. Following the (Kim et al., 2023)-style protocol, we define a small set of **anchor** programs that are always included in training to ensure basic coverage of the underlying space. In GridWorld, the anchors are the four "triple-move" programs that apply the same motion primitive three times (i.e., UUU, DDD, LLL, RRR); these anchors provide coarse coverage such that other short displacements can, in theory, be obtained via interpolation within the short-program space. For each $\alpha$, we then sample an $\alpha$ fraction of the remaining short programs into training, and hold out the rest as **Compositional OOD**. For evaluation, we collect 5,000 compositional-OOD instances per $\alpha$, and a separate length-OOD set of 10,000 instances (shared across $\alpha$-splits) generated from 4–8 step programs.

**Max Transition Length $K$.**   For the GridWorld task, the training data includes compositions of up to three primitives, and we set the max transition length to $K = 4$ during training. The same value is used for in-distribution and compositional OOD inference. For length OOD evaluation, where transformations involve compositions of up to eight primitives, we increase the max transition length to $K = 10$ to allow the model to construct longer explanations.

**(a)** $\alpha = 0.33$

**Training:** Anchors) `UUU, DDD, LLL, RRR`. Randomly sampled) `U, LU, DL, DR, DD, DDL, DRR`.

**Compositional OOD: `L, R, D,` `LL, RR, UU, RU, RRL, LLU, DLL, RUU, DDR, RRU`.**

**(b)** $\alpha = 0.66$

**Training:** Anchors: `UUU, DDD, LLL, RRR`. Randomly sampled: `U, D, R, LU, DL, DR, UU, DD, RR, LUU, LLU, DRR, DDR, RRU`.

**Compositional OOD: `L,` `LL, DLL, RU, RUU, DDR`.**

**(c)** $\alpha = 1.00$

**Training:** Anchors: `UUU, DDD, LLL, RRR`. All remaining programs are included.

**Compositional OOD:** None.

### C.4. GridWorld Results

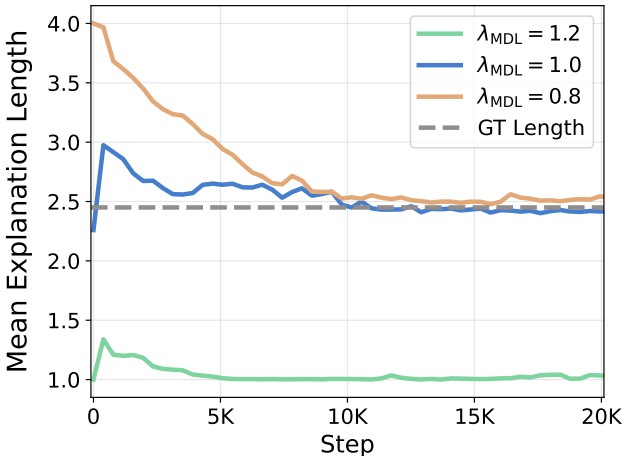

*Figure 8.* **Mean explanation length over training for different MDL weights** $\lambda_{\mathrm{MDL}}$. Larger $\lambda_{\mathrm{MDL}}$ encourages shorter explanations, while smaller $\lambda_{\mathrm{MDL}}$ yields longer explanations

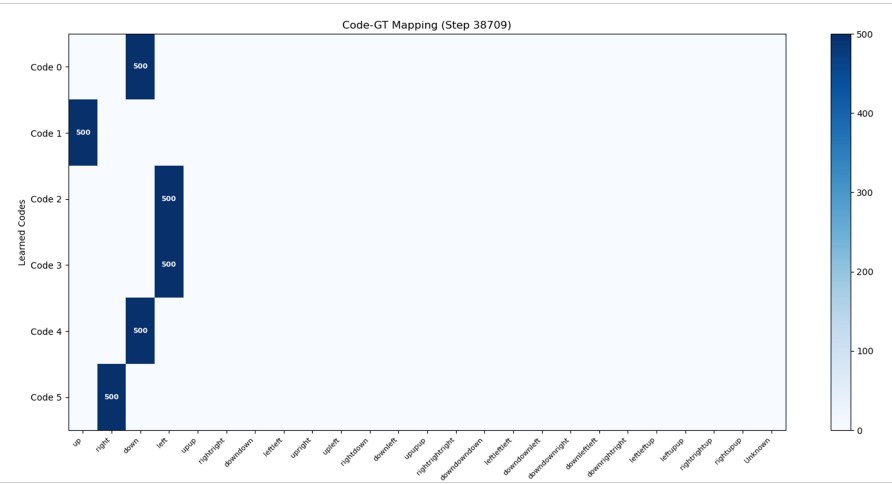

*Figure 9.* Code–primitive alignment in GridWorld $\alpha = 0.33$ ($|\mathcal{E}| = 6$). Each row is a learned code and each column is a ground-truth primitive transformation; counts indicate how often a code is assigned to a primitive. The near one-to-one structure shows that the codebook captures primitive-level actions rather than entangled programs.

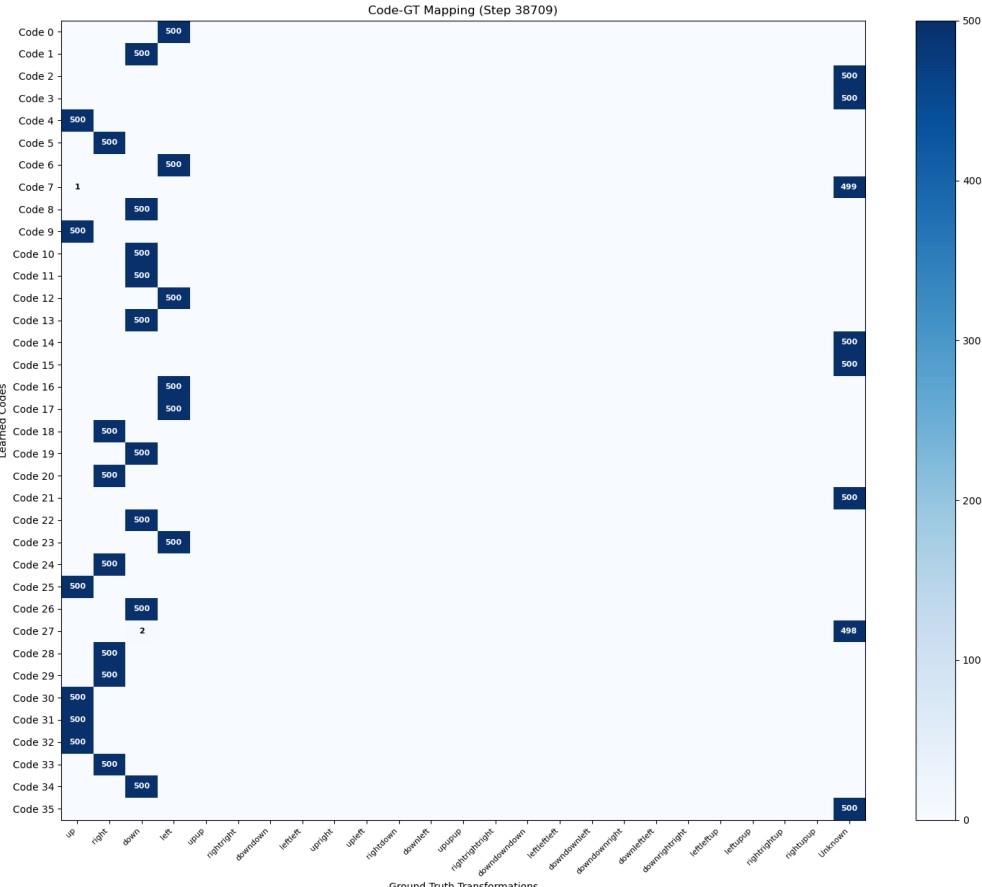

*Figure 10.* Code–primitive alignment in GridWorld $\alpha = 0.33$ ($\lambda_{\mathrm{MDL}} = 0.8$). Each row is a learned code and each column is a ground-truth primitive transformation; counts indicate how often a code is assigned to a primitive. The codebook captures primitive-level actions rather than entangled programs.

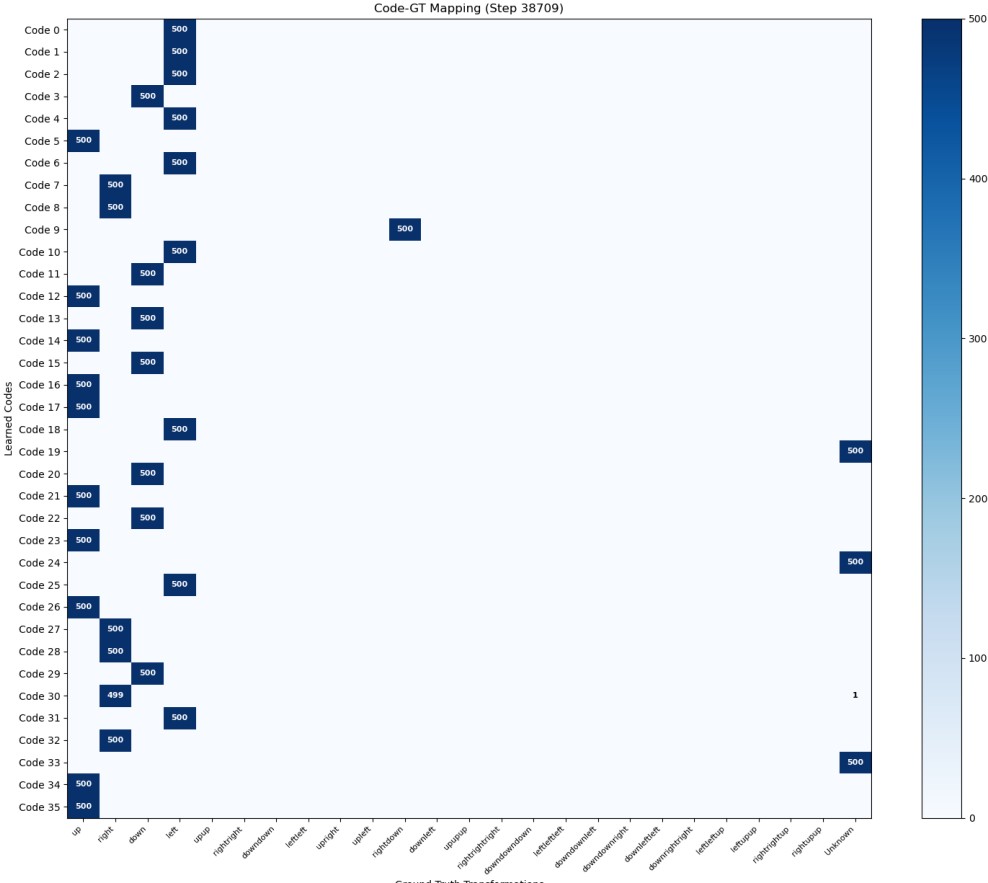

*Figure 11.* Code–primitive alignment in GridWorld $\alpha = 0.33$ ($\lambda_{\mathrm{MDL}} = 1.0$). Most learned codes align with the four ground-truth motion primitives, indicating successful primitive recovery. Interestingly, a small number of codes capture short composite motions (e.g., *right–down*), suggesting that with a slightly weaker pressure toward multi-step decomposition, the codebook can also allocate capacity to frequent entangled subroutines while largely preserving primitive-level structure.

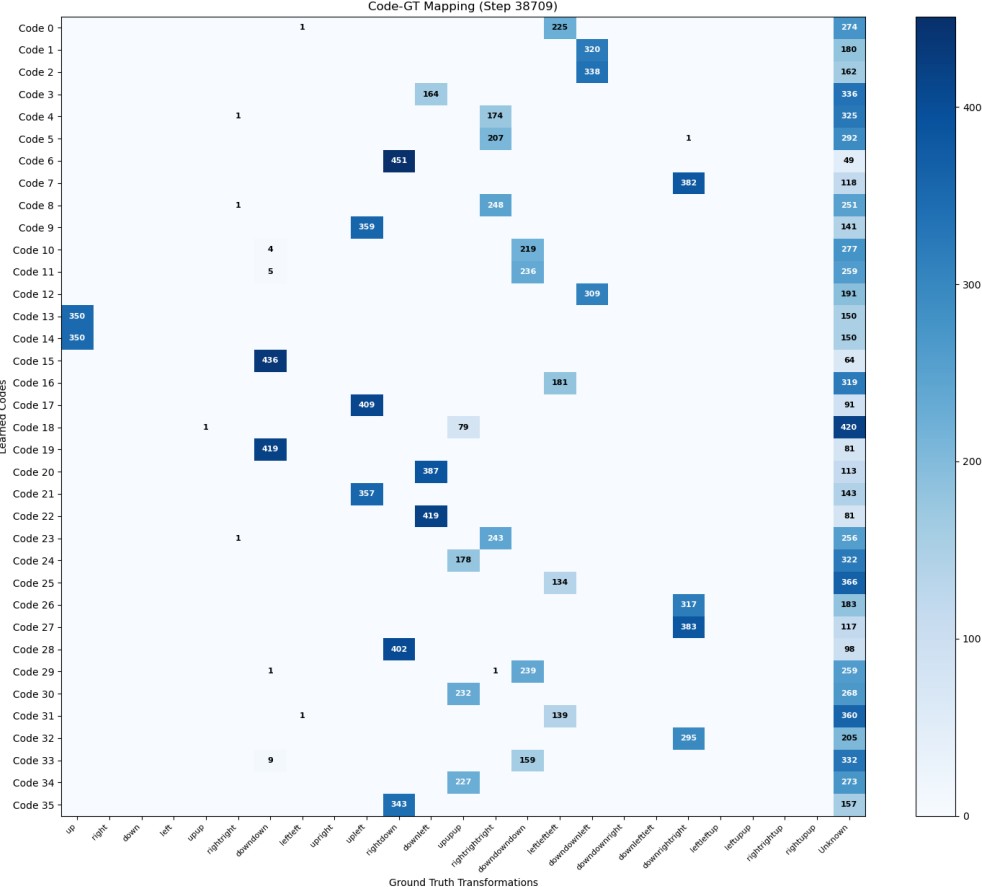

*Figure 12.* Code–primitive alignment in GridWorld $\alpha = 0.33$ ($\lambda_{\mathrm{MDL}} = 1.2$). In contrast to smaller $\lambda_{\mathrm{MDL}}$, the mapping no longer exhibits a near alignment with the four ground-truth motion primitives. Instead, many codes specialize to *composite* (entangled) transformations, indicating that a larger $\lambda_{\mathrm{MDL}}$ shifts learning toward memorizing short programs rather than recovering primitive-level actions.

## C.5. Arithmetic Factorization Reasoning Details

### C.5.1. ARITHMETIC FACTORIZATION REASONING PRIMITIVES

*Table 5.* Ground-truth primitives used in the Arithmetic Factorization Reasoning task.

| Primitive | Description |
|---|---|
| $\times 2$ | Multiply 2 to current number. |
| $\times 3$ | Multiply 3 to current number. |
| $\times 5$ | Multiply 5 to current number. |
| $\times 7$ | Multiply 7 to current number. |

### C.5.2. ARITHMETIC FACTORIZATION REASONING TEST SETTING

**Dataset Construction.** We construct OTIB-Arithmetic with *short* programs of length 1–3 primitive steps, and evaluate **length OOD** on *longer* programs of length 4–6 steps. Following the SVIB ([Kim et al., 2023](#))-style protocol, we define **anchor** programs that are always included in training: the 3 repeated-multiplication sequences ($\times 2 \times 2 \times 2$, $\times 3 \times 3 \times 3$, $\times 5 \times 5 \times 5$, $\times 7 \times 7 \times 7$). Unlike OTIB-GridWorld, decomposing repeated multiplication into individual primitives is non-trivial. For each $\alpha$, we sample an $\alpha$ fraction of the remaining short programs into training and hold out the rest as **compositional OOD**, stratified by program length. We evaluate on 1,000 compositional-OOD instances per program combination per $\alpha$, and 5,000 length-OOD instances (shared across $\alpha$-splits) generated from 4–6 step programs. We report held-out combinations for compositional OOD test in Table 6.

**Max Transition Length $K$.** For the Arithmetic Factorization Reasoning task, the training data includes compositions of up to three primitives, and we set the max transition length to $K = 3$ during training. The same value is used for in-distribution and compositional OOD inference. For length OOD evaluation, where transformations involve compositions of up to 6 primitives, we increase the max transition length to $K = 6$ to allow the model to construct longer explanations.

*Table 6.* Held-out programs for compositional OOD evaluation in OTIB-Arithmetic. Bold programs indicate single primitives that never appear in isolation during training.

| $\alpha$ | **Held-out Programs** |
|---|---|
| 0.33 | $\times\mathbf{3}$, $\times\mathbf{5}$, $\times 2 \times 5$, $\times 2 \times 7$, $\times 3 \times 5$, $\times 3 \times 7$, $\times 2 \times 3^2$, $\times 2^2 \times 5$, $\times 2^2 \times 7$, $\times 2 \times 3 \times 7$, $\times 3^2 \times 5$, $\times 2 \times 5^2$, $\times 2 \times 7^2$, $\times 3 \times 5 \times 7$, $\times 3 \times 7^2$, $\times 5 \times 7^2$ |
| 0.66 | $\times\mathbf{3}$, $\times 3 \times 5$, $\times 2 \times 5$, $\times 3 \times 7^2$, $\times 2^2 \times 7$, $\times 3^2 \times 5$, $\times 2 \times 3 \times 7$, $\times 2^2 \times 5$ |
| 1.00 | None |

### C.5.3. ARITHMETIC FACTORIZATION REASONING RESULTS

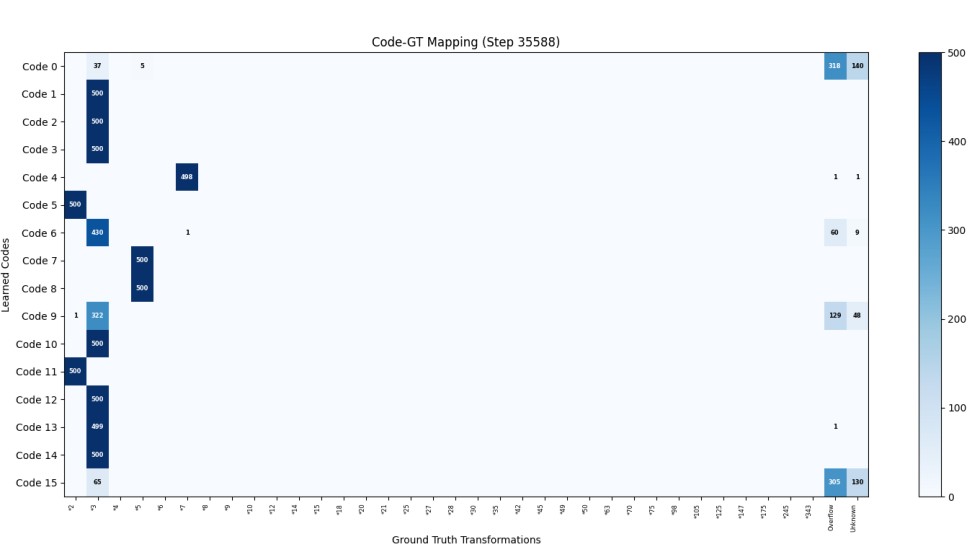

*Figure 13.* Code–primitive alignment in Arithmetic Factorization Task $\alpha = 0.33$ ($|\mathcal{E}| = 16$). Despite being given an overcomplete codebook, NEO discovers and utilizes only the true underlying primitives, demonstrating that the model learns to identify the minimal set of reusable operations rather than exploiting excess capacity.

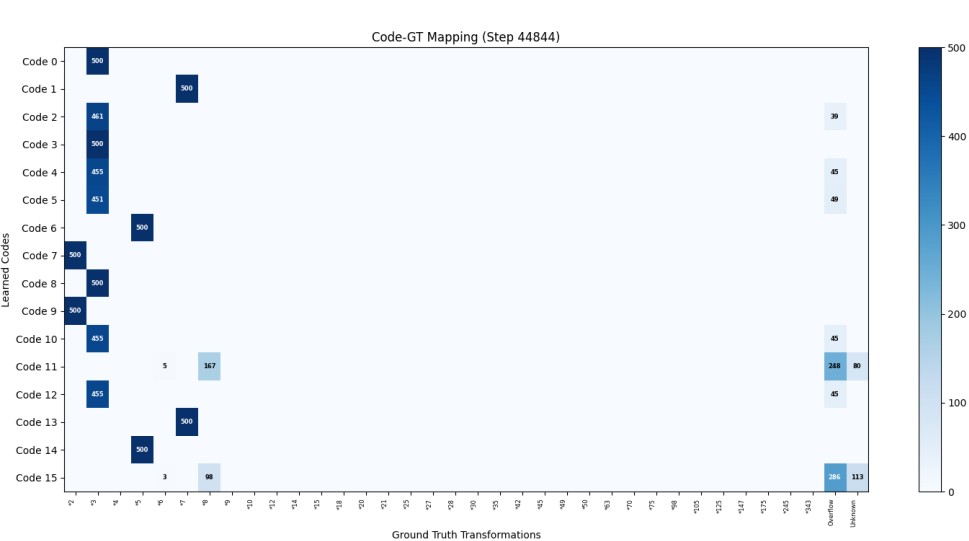

*Figure 14.* Code–primitive alignment in Arithmetic Factorization Task $\alpha = 0.66$ ($|\mathcal{E}| = 16$). Even with an overcomplete codebook, NEO learns to use only the true underlying primitives, identifying the minimal set of reusable operations rather than exploiting excess capacity.

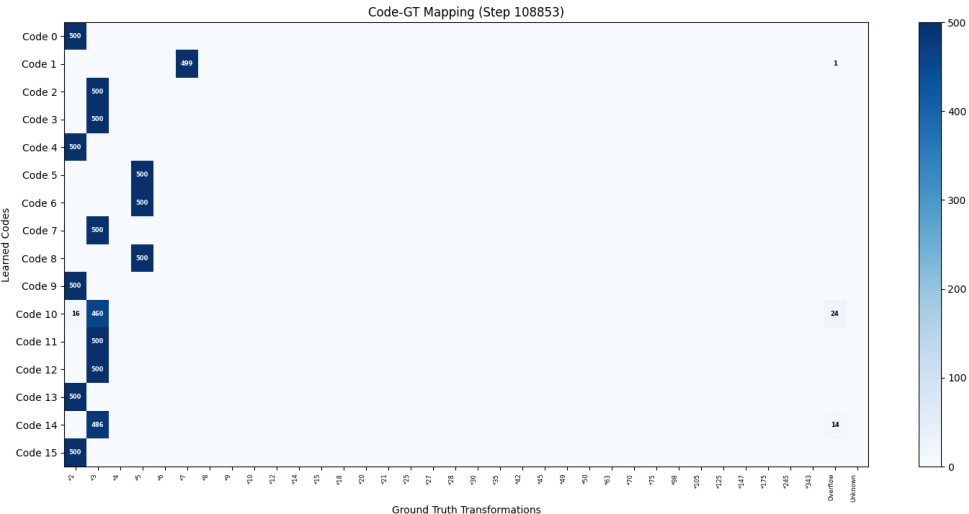

*Figure 15.* Code–primitive alignment in Arithmetic Factorization Task $\alpha = 1.00$ ($|\mathcal{E}| = 16$).

## C.6. Image Editing Details

### C.6.1. IMAGE EDITING PRIMITIVES

*Table 7.* Ground-truth image editing primitives used in the Image Editing task.

| Category | Primitive | Description |
|---|---|---|
| Brightness | `brightness_plus(br_p)` | Increase brightness by a factor of 1.5 |
| | `brightness_minus (br_m)` | Decrease brightness by a factor of 0.5 |
| Hue | `hue_plus(hue_p)` | Rotate hue by +0.3 on the color wheel |
| | `hue_minus(hue_m)` | Rotate hue by $-0.3$ on the color wheel |
| Flip | `horizontal_flip(h_flip)` | Flip image horizontally (left–right) |
| | `vertical_flip(v_flip)` | Flip image vertically (top–bottom) |
| Others | `rotation(rot)` | Rotate image by $90°$ clockwise |
| | `masking(mask)` | Apply a gray (128) square mask to the top-left quadrant (25% of image size) |

### C.6.2. IMAGE EDITING TEST SETTING

**Dataset Construction.** The image editing dataset is constructed based on CIFAR-10 (Krizhevsky & Hinton, 2009), following the official train–test split. Each sample is generated by applying one or more editing primitives to an input image, and only transformations whose pixel-wise difference exceeds a predefined threshold are retained to ensure semantic relevance. We consider a total of 8 primitives, summarized in Table 7. Action sequences are composed under a canonical ordering to avoid redundant permutations.

IID and OOD settings are defined by the set of primitive compositions observed during training. In the IID and OOD splits, sequences of length one or two are used, while the Length-OOD setting consists of longer compositions of three to four primitives. The parameter $\alpha$ controls the fraction of compositions included in the IID set, with smaller $\alpha$ inducing more severe distribution shifts. Only canonical compositions that produce sufficiently large transformations are included. All IID and OOD composition cases are listed in Tables 8, 9, and 10 for $\alpha = 0.33, 0.66$, and $1.0$, respectively.

*Table 8.* IID and OOD combinations for $\alpha = 0.33$.

| Split | Category | Combinations |
|---|---|---|
| IID | Level 1 | `[horizontal_flip]`, `[masking]`, `[vertical_flip]` |
| | Level 2 | `[br_m, br_m]`, `[br_p, br_p]`, `[hue_m, hue_m]`, `[hue_p, hue_p]`, `[rot, rot]`, `[br_m, hue_m]`, `[br_m, hue_p]`, `[br_m, mask]`, `[br_m, rot]`, `[br_m, v_flip]`, `[br_p, hue_m]`, `[br_p, hue_p]`, `[h_flip, rot]`, `[hue_p, mask]`, `[rot, v_flip]` |
| OOD | Level 1 | `brightness_minus`, `brightness_plus`, `hue_minus`, `hue_plus`, `rotation` |
| | Level 2 | `[br_m, h_flip]`, `[br_p, h_flip]`, `[br_p, mask]`, `[br_p, rot]`, `[br_p, v_flip]`, `[h_flip, hue_m]`, `[h_flip, hue_p]`, `[h_flip, mask]`, `[h_flip, v_flip]`, `[hue_m, mask]`, `[hue_m, rot]`, `[hue_m, v_flip]`, `[hue_p, rot]`, `[hue_p, v_flip]`, `[mask, rot]`, `[mask, v_flip]` |

*Table 9.* IID and OOD combinations for $\alpha = 0.66$.

| Split | Category | Combinations |
|-------|----------|--------------|
| IID | Level 1 | `[horizontal_flip]`, `[masking]`, `[vertical_flip]`, `[brightness_plus]`, `[hue_plus]` |
|  | Level 2 | `[br_m, br_m]`, `[br_p, br_p]`, `[hue_m, hue_m]`, `[hue_p, hue_p]`, `[rot, rot]`, `[br_m, h_flip]`, `[br_m, hue_m]`, `[br_m, hue_p]`, `[br_m, mask]`, `[br_m, rot]`, `[br_m, v_flip]`, `[br_p, h_flip]`, `[br_p, hue_m]`, `[br_p, hue_p]`, `[br_p, rot]`, `[h_flip, hue_p]`, `[h_flip, rot]`, `[hue_m, mask]`, `[hue_m, rot]`, `[hue_m, v_flip]`, `[hue_p, mask]`, `[hue_p, v_flip]`, `[rot, v_flip]` |
| OOD | Level 1 | `[brightness_minus]`, `[hue_minus]`, `[rotation]` |
|  | Level 2 | `[br_p, mask]`, `[br_p, v_flip]`, `[h_flip, hue_m]`, `[h_flip, mask]`, `[h_flip, v_flip]`, `[hue_p, rot]`, `[mask, rot]`, `[mask, v_flip]` |

*Table 10.* Full list of IID combinations for $\alpha = 1.0$.

| Category | Combinations |
|----------|--------------|
| Level 1 (IID) | `[rotation]`, `[horizontal_flip]`, `[masking]`, `[vertical_flip]`, `[brightness_minus]`, `[brightness_plus]`, `[hue_minus]`, `[hue_plus]` |
| Level 2 (IID) | `[br_m, br_m]`, `[br_p, br_p]`, `[hue_m, hue_m]`, `[hue_p, hue_p]`, `[rot, rot]`, `[br_m, hue_m]`, `[br_m, hue_p]`, `[br_m, mask]`, `[br_m, rot]`, `[br_m, v_flip]`, `[br_p, hue_m]`, `[br_p, hue_p]`, `[h_flip, rot]`, `[hue_p, mask]`, `[rot, v_flip]`, `[br_m, h_flip]`, `[br_p, h_flip]`, `[br_p, mask]`, `[br_p, rot]`, `[br_p, v_flip]`, `[h_flip, hue_m]`, `[h_flip, hue_p]`, `[h_flip, mask]`, `[h_flip, v_flip]`, `[hue_m, mask]`, `[hue_m, rot]`, `[hue_m, v_flip]`, `[hue_p, rot]`, `[hue_p, v_flip]`, `[mask, rot]`, `[mask, v_flip]` |

**Length-OOD Setting.** The Length-OOD dataset consists of compositions of three to four primitives, which are never observed during training. After canonicalization, we obtain 79 valid compositions for length three (from 512 permutations) and 152 for length four (from 4096 permutations), yielding a total of 231 valid combinations. As in the IID/OOD splits, only transformations exceeding the difference threshold are retained.

**Max Transition Length $K$.** For the Image Editing task, training data consists of compositions of one or two primitives, and we therefore set $K = 3$ during training. The same setting is used for in-distribution and compositional OOD inference. For length OOD evaluation, where transformations involve compositions of three or four primitives, we increase the max transition length to $K = 6$ to accommodate longer explanations.

### C.7. Image Editing Results

#### C.7.1. FULL PERFORMANCE COMPARISON.

As shown in Table 11, our method consistently outperforms the baselines across most evaluation settings, not only in self-explainability but also in transfer performance, particularly under out-of-distribution conditions.

For reference, a trivial identity baseline (L1 distance between $x$ and $y$) yields 0.212 under compositional OOD and 0.271 under length OOD, whereas NEO achieves substantially lower errors in the range of 0.09–0.12.

*Table 11.* **Performance comparison on the Image Editing environment.** Results show mean across three runs for each metric.

| $\alpha$ | Method | In-distribution | | Comp. OOD | | Length OOD | |
|---|---|---|---|---|---|---|---|
| | | Self-Ex. | Transf. | Self-Ex. | Transf. | Self-Ex. | Transf. |
| 0.33 | Disc-Mono | 0.06 | **0.06** | 0.16 | 0.17 | 0.18 | 0.19 |
| | Cont-Mono | **0.05** | 0.08 | 0.14 | 0.17 | 0.16 | 0.19 |
| | Cont-Mono-Opt | **0.05** | 0.08 | 0.13 | 0.16 | 0.16 | 0.19 |
| | **NEO (Ours)** | 0.06 | 0.07 | **0.11** | **0.12** | **0.12** | **0.13** |
| 0.66 | Disc-Mono | 0.07 | **0.07** | 0.14 | 0.15 | 0.16 | 0.17 |
| | Cont-Mono | **0.06** | 0.13 | 0.13 | 0.18 | 0.15 | 0.21 |
| | Cont-Mono-Opt | **0.06** | 0.13 | 0.12 | 0.18 | 0.14 | 0.21 |
| | **NEO (Ours)** | 0.07 | **0.07** | **0.09** | **0.09** | **0.11** | **0.11** |
| 1.00 | Disc-Mono | 0.08 | 0.08 | · | · | 0.16 | 0.17 |
| | Cont-Mono | **0.06** | 0.12 | · | · | 0.13 | 0.18 |
| | Cont-Mono-Opt | **0.06** | 0.12 | · | · | 0.12 | 0.18 |
| | **NEO (Ours)** | 0.07 | **0.07** | · | · | **0.10** | **0.10** |

## C.7.2. CODE–PRIMITIVE ALIGNMENT MATRIX.

For the Image Editing experiments, we construct a code–primitive alignment matrix by selecting, for each code, the ground truth primitive action with the lowest reconstruction loss. In addition to comparing against all action primitives present in the dataset, we include a no-operation primitive corresponding to the original image. This allows the confusion matrix to capture both meaningful action alignments and cases where a code represents identity-preserving transformations.

**(a) $\alpha = 0.33$.** As shown in Figure 16, when $\alpha = 0.33$, the model successfully discovers three distinct primitive codes, despite five primitive operators being entirely absent from the IID dataset. This demonstrates the model's ability to induce meaningful primitives from limited and incomplete supervision

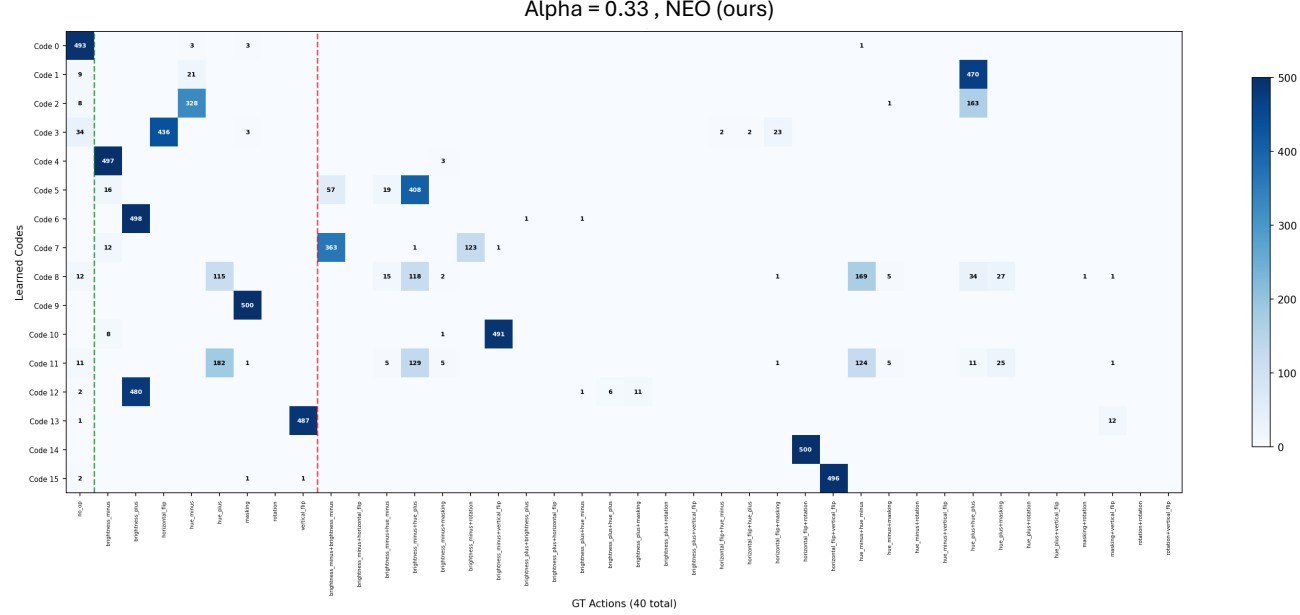

*Figure 16.* Code–primitive alignment in Image Editing $\alpha = 0.33$.

**(b) $\alpha = 0.66$.** As shown in Figure 17, when $\alpha = 0.66$, the model correctly recovers all three missing primitive operators that are not directly observed in the IID dataset. This highlights its capacity to decompose composed transformations and to identify reusable, general-purpose action primitives underlying observed phenomena.

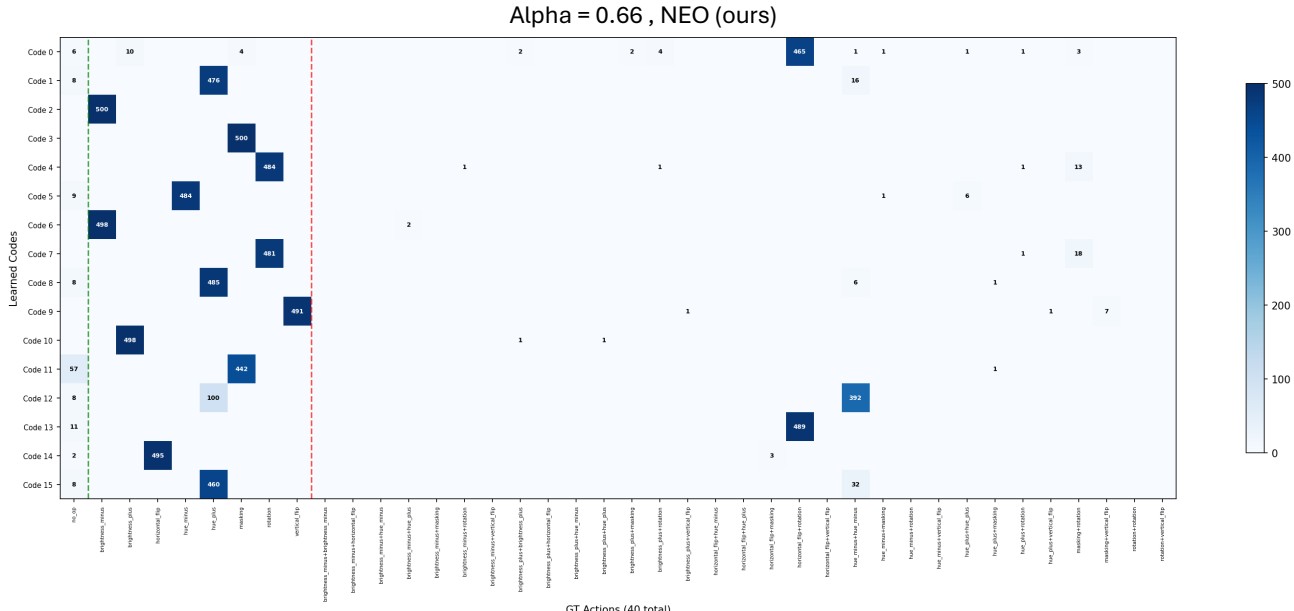

*Figure 17.* Code–primitive alignment in Image Editing $\alpha = 0.66$.

**(c) $\alpha = 1.0$.** As shown in Figure 18, when $\alpha = 1.0$, the model consistently recovers all primitive codes, even though all actions are uniformly represented in the dataset. Notably, the learned codebook aligns with the most elementary action primitives, indicating a preference for minimal and fundamental explanations.

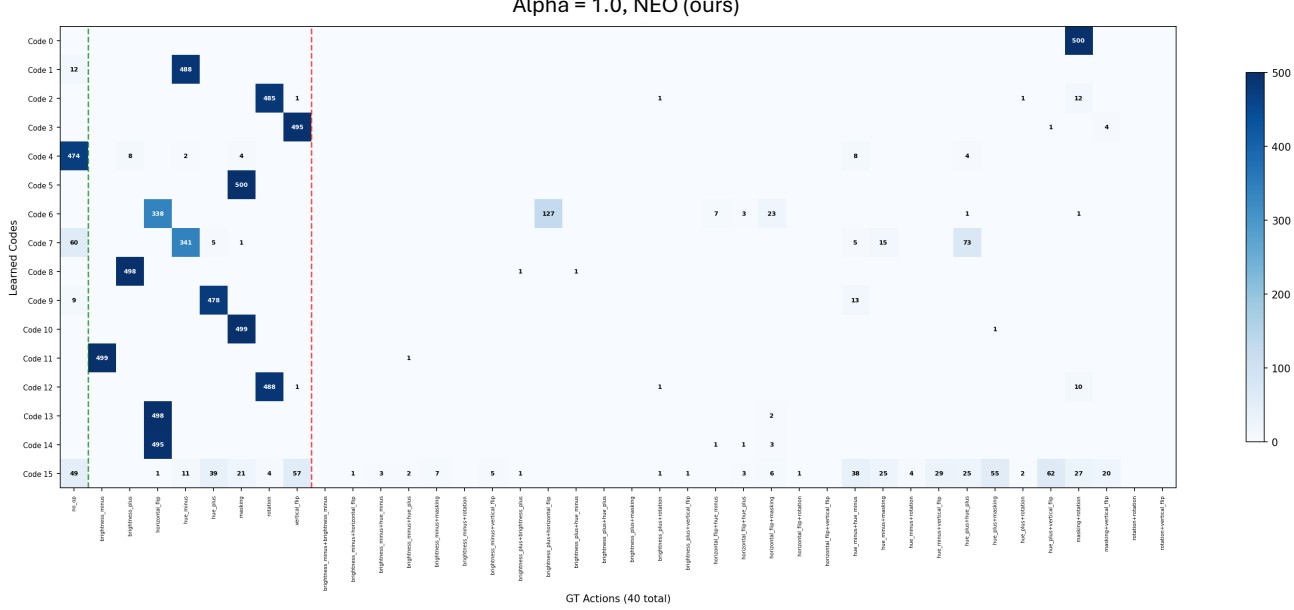

*Figure 18.* Code–primitive alignment in Image Editing $\alpha = 1.0$.

# D. Additional Analysis

## D.1. Test Time Scaling

**GridWorld**    We conduct test-time scaling on GridWorld by sampling $B \in \{1, 2, 4, 8, 16, 32, 64\}$ candidate theories from the probabilistic theory programmer and selecting a single theory via majority voting before transfer. We report both self-explainability and transferability in Figure 19; importantly, majority voting is performed over theories induced from the same observation $x \rightarrow y$ and does not use any additional supervision.

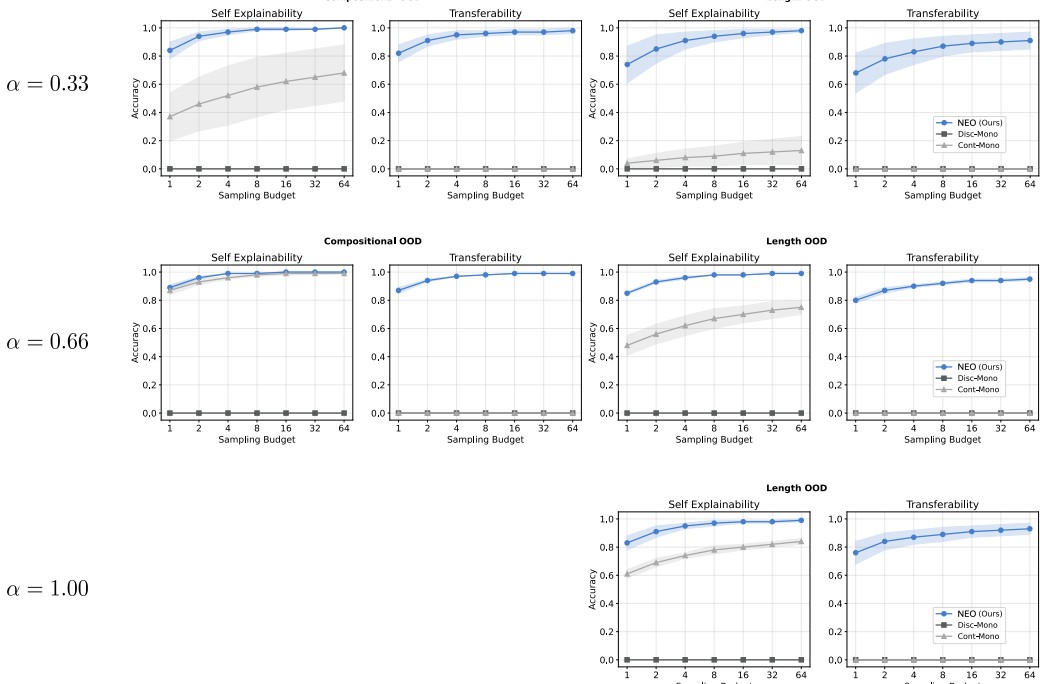

*Figure 19.* Test-time scaling results on GridWorld domain.

## D.2. Arithmetic Factorization Reasoning

**Arithmetic Factorization Reasoning**    We conduct test-time scaling on Arithmetic Factorization Reasoning by sampling $B \in \{1, 4, 16, 64, 256, 1024\}$ candidate theories from the probabilistic theory programmer and selecting a single theory via majority voting before transfer. We report transferability in Figure 20; importantly, majority voting is performed over theories induced from the same observation $x \rightarrow y$ and does not use any additional supervision.

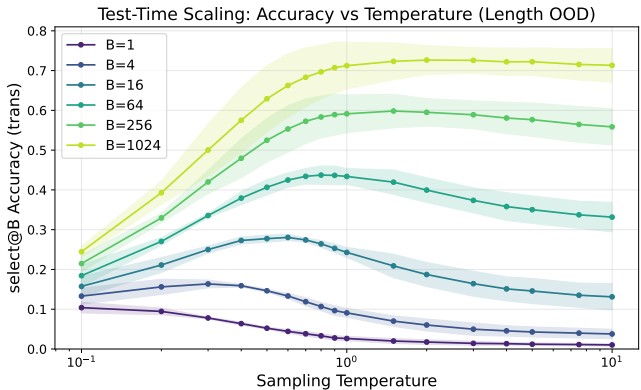

*Figure 20.* **Test-time scaling on Arithmetic Reasoning (Length OOD).** Transfer accuracy improves with both sampling budget $B$ and temperature, demonstrating that NEO's compositional structure enables effective test-time scaling. Higher temperatures encourage exploration of diverse primitive compositions, while larger budgets increase the probability of finding correct programs.

### D.3. Computational Resource Analysis

**Computational Cost.** We note that theory induction prioritizes the quality of the discovered explanation over inference speed. Nevertheless, computational efficiency remains an important practical consideration, and we provide a detailed comparison across methods. **Disc-Mono / Cont-Mono** perform a single forward pass. **Cont-Mono-Opt** additionally performs iterative gradient-based optimization of the program vector at inference time. **NEO** performs $K$ sequential forward passes through the transition network. **NEO-S** repeats the $K$-step rollout $B$ times with different sampled programs, scaling as $\mathcal{O}(K \times B)$. Within each domain, all methods share the same pretrained encoder and decoder.

**GridWorld (NVIDIA RTX 3090, batch size 128).**

*Table 12.* Computational cost comparison on GridWorld.

| Method | Train (ms/batch) | Total Train Time (s) | Inference (ms/batch) |
|---|---|---|---|
| Disc-Mono | 49.8 | 5,835 (150 ep) | 28.3 |
| Cont-Mono | 50.2 | 5,882 (150 ep) | 26.6 |
| Cont-Mono-Opt | – | – | 91.2 |
| NEO (Ours) | 75.4 | 5,890 (100 ep) | 41.1 |
| NEO-S ($B = 8$) | – | – | 42.5 |
| NEO-S ($B = 64$) | – | – | 174.0 |

**Arithmetic Factorization Reasoning (NVIDIA RTX 4090, batch size 512).**

*Table 13.* Computational cost comparison on arithmetic reasoning.

| Method | Train (ms/batch) | Total Train Time (s) | Inference (ms/batch) |
|---|---|---|---|
| Disc-Mono | 25.0 | 9,920 (200 ep) | 13.0 |
| Cont-Mono | 23.1 | 9,900 (200 ep) | 12.1 |
| Cont-Mono-Opt | – | – | 178.9 |
| NEO (Ours) | 78.9 | 14,300 (200 ep) | 27.5 |
| NEO-S ($B = 8$) | – | – | 42.1 |
| NEO-S ($B = 1024$) | – | – | 4867.3 |

**Image Editing (NVIDIA RTX 4090, batch size 64).**

*Table 14.* Computational cost comparison on image editing.

| Method | Train (ms/batch) | Total Train Time (s) | Inference (ms/batch) |
|---|---|---|---|
| Disc-Mono | 58.7 | 26,816 (50 ep) | 24.53 |
| Cont-Mono | 50.1 | 22,925 (50 ep) | 23.41 |
| Cont-Mono-Opt | – | – | 571.35 |
| NEO (Ours) | 106.7 | 48,779 (50 ep) | 38.06 |

Overall, NEO incurs approximately $2\times$ higher training cost than single-pass baselines due to sequential transitions. At inference time, it is slower than single-pass methods but significantly more efficient than gradient-based optimization (Cont-Mono-Opt). For NEO-S, the additional cost from repeated sampling directly trades off with improved out-of-distribution generalization.

**Dataset Sizes.** We report the number of training and evaluation examples across domains, including in-distribution (ID), compositional OOD, and length OOD settings.

**GridWorld.**

*Table 15.* Dataset sizes for GridWorld.

| Setting | Train | ID Test | Comp. OOD Test | Length OOD Test |
|---|---|---|---|---|
| $\alpha = 1.00$ | 100,000 | 10,000 | – | 20,000 |
| $\alpha = 0.66$ | 100,000 | 10,000 | 10,000 | 20,000 |
| $\alpha = 0.33$ | 100,000 | 10,000 | 10,000 | 20,000 |

**Arithmetic Factorization Reasoning.**

*Table 16.* Dataset sizes for arithmetic reasoning.

| Setting | Train | ID Test | Comp. OOD Test | Length OOD Test |
|---|---|---|---|---|
| $\alpha = 1.00$ | 279,611 | 27,961 | – | 15,317 |
| $\alpha = 0.66$ | 206,520 | 20,651 | 80,401 | 15,317 |
| $\alpha = 0.33$ | 147,968 | 14,796 | 146,306 | 15,317 |

**Image Editing.**

*Table 17.* Dataset sizes for image editing.

| Setting | Train | ID Test | Comp. OOD Test | Length OOD Test |
|---|---|---|---|---|
| $\alpha = 1.00$ | 1,170,000 | 234,000 | – | 393,886 |
| $\alpha = 0.66$ | 1,120,000 | 224,000 | 440,000 | 393,886 |
| $\alpha = 0.33$ | 720,000 | 144,000 | 840,000 | 393,886 |

# E. Visualization Results

In this section, we visualize the model's rollout results to qualitatively analyze NEO's behavior.

### E.1. GridWorld

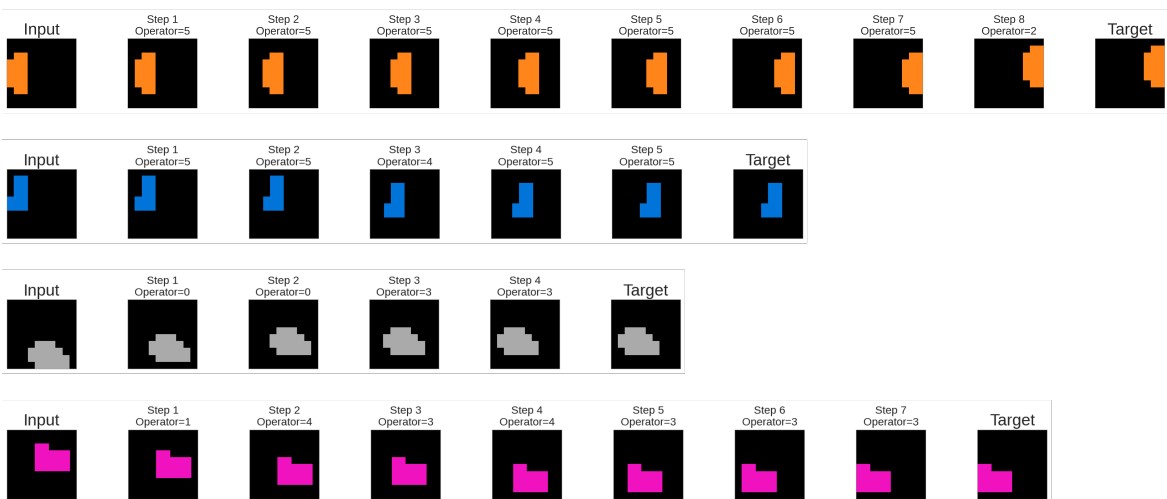

*Figure 21.* NEO visualization on length OOD task.

### E.2. Arithmetic Factorization Reasoning

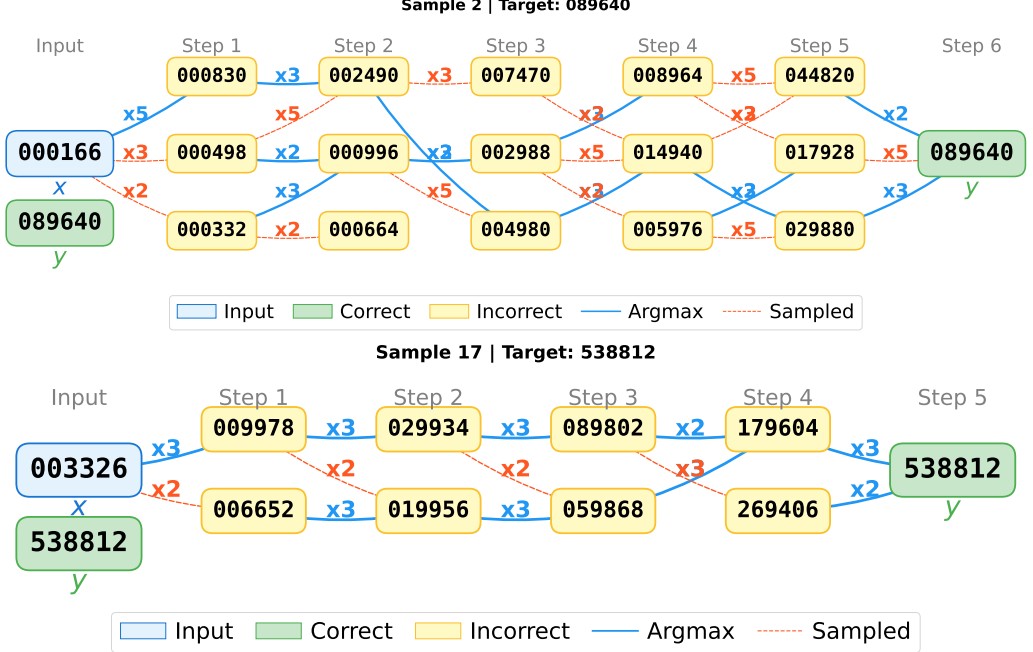

*Figure 22.* NEO visualization on length OOD task. Sampled with budget $B = 1024$ and temperature $\tau = 1.0$.

# F. Model Hyperparameters

For reproducibility, we report the hyperparameters used in all experiments. The settings were guided by prior work and empirical validation to ensure stable training and consistent evaluation. We use largely shared hyperparameters across tasks, introducing task-specific configurations only when necessary, as detailed in the respective sections.

We briefly describe the pretraining setup used to define the latent space for our method. For GridWorld and Image Editing, we employ a CNN-based variational autoencoder (VAE) to learn latent representations prior to training. In contrast, for Arithmetic Reasoning, we directly learn embedding vectors and perform all operations within the resulting latent space. Detailed hyperparameter configurations for pretraining and reconstruction are provided below.

For Arithmetic Reasoning and Image Editing, we apply a linear scheduling of the MDL length penalty $\lambda_{\text{MDL}}$ over the course of training, enabled when use_$\lambda_{\text{MDL}}$_scheduling is set to true. Specifically, $\lambda_{\text{MDL}}$ is annealed linearly from $\lambda_{\text{MDL}}^{\text{start}}$ to $\lambda_{\text{MDL}}^{\text{end}}$. This design encourages the model to first discover and stabilize primitive operations during early training, and subsequently to compose them into explanations whose length adapts to the underlying complexity of the transformation.

## F.1. GridWorld

*Table 18.* Pretraining hyperparameters for the state VAE in GridWorld experiments. The same pretrained model is used for NEO and all baselines.

| Hyperparameter | Value |
| --- | --- |
| *Training* | |
| Learning Rate | $5e{-}3$ |
| Weight Decay | $1e{-}2$ |
| LR Scheduler | Cosine Annealing |
| Min LR Ratio | 0.005 |
| Precision | bf16-mixed |
| Batch Size | 512 |
| Max Epochs | 500 |
| Gradient Clipping | 1.0 |
| *Architecture* | |
| State Encoder | CNN |
| State Decoder | CNN |
| Hidden Dimension ($d_{\text{model}}$) | 32 |
| Feedforward Dimension ($d_{\text{ff}}$) | 128 |
| Dropout | 0.1 |
| State Dimension | 32 |
| *VAE* | |
| VAE $\beta$ | $1e{-}5$ |
| *Loss Weights* | |
| Auto Reconstruction Loss | 1.0 |

*Table 19.* GridWorld $\alpha = 0.33$ Experiments Hyperparameters. We conducted experiments on three seed values.

| Hyperparameter | NEO | Disc-Mono | Cont-Mono | Cont-Mono-Opt |
|---|---|---|---|---|
| *Training* | | | | |
| Learning Rate | $5e{-}4$ | $5e{-}3$ | $5e{-}3$ | $5e{-}3$ |
| Weight Decay | $1e{-}2$ | $1e{-}2$ | $1e{-}2$ | $1e{-}2$ |
| Warmup Ratio | 0.1 | 0.05 | 0.05 | 0.05 |
| LR Scheduler | Cosine | Cosine | Cosine | Cosine |
| Min LR Ratio | 0.1 | 0.1 | 0.1 | 0.1 |
| Precision | bf16-mixed | bf16-mixed | bf16-mixed | bf16-mixed |
| Batch Size | 128 | 128 | 128 | 128 |
| Max Epochs | 100 | 150 | 100 | 100 |
| Gradient Clipping | 1.0 | 1.0 | 1.0 | 1.0 |
| *Two-Timescale Learning Rate* | | | | |
| Policy LR Scale | 0.25 | 0.25 | 0.25 | 0.25 |
| Transition LR Scale | 1.0 | 1.0 | 1.0 | 1.0 |
| *Architecture* | | | | |
| State Encoder | CNN | CNN | CNN | CNN |
| State Decoder | CNN | CNN | CNN | CNN |
| Policy Network | FiLM-MLP | FiLM-MLP | FiLM-MLP | FiLM-MLP |
| Transition Network | FiLM-MLP | FiLM-MLP | FiLM-MLP | FiLM-MLP |
| Hidden Dimension ($d_{\text{model}}$) | 32 | 32 | 32 | 32 |
| Feedforward Dimension ($d_{\text{ff}}$) | 128 | 128 | 128 | 128 |
| State Dimension | 32 | 32 | 32 | 32 |
| Action Dimension | 16 | 16 | 16 | 16 |
| Num Action Tokens | 1 | 1 | 1 | 1 |
| Max Transition Length (K) | 4 | 1 | 1 | 1 |
| *Latent Action Space* | | | | |
| Action Space Type | Discrete (VQ) | Discrete (VQ) | Continuous (VAE) | Continuous (VAE) |
| Codebook Size | 6 | 36 | – | – |
| Commitment Cost ($\beta$) | 0.25 | 0.25 | – | – |
| Gumbel-Softmax $\tau_{\text{start}}$ | 0.3 | 0.3 | – | – |
| Gumbel-Softmax $\tau_{\text{end}}$ | 0.1 | 0.1 | – | – |
| VAE $\beta$ | – | – | 0.01 | 0.01 |
| *Loss Weights* | | | | |
| Reconstruction Loss | 1.0 | 1.0 | 1.0 | 1.0 |
| Grounding Loss | 0.1 | – | – | – |
| Action VQ Loss | 1.0 | 1.0 | – | – |
| $\lambda_{\text{MDL}}$ | 0.95 | – | – | – |
| *Test-Time Optimization* | | | | |
| Gradient Search Steps | – | – | – | 5 |

*Table 20.* GridWorld $\alpha = 0.66$ Experiments Hyperparameters. We conducted experiments on three seed values.

| Hyperparameter | NEO | Disc-Mono | Cont-Mono | Cont-Mono-Opt |
|---|---|---|---|---|
| *Training* | | | | |
| Learning Rate | $5e{-}4$ | $5e{-}3$ | $5e{-}3$ | $5e{-}3$ |
| Weight Decay | $1e{-}2$ | $1e{-}2$ | $1e{-}2$ | $1e{-}2$ |
| Warmup Ratio | 0.05 | 0.05 | 0.05 | 0.05 |
| LR Scheduler | Cosine | Cosine | Cosine | Cosine |
| Min LR Ratio | 0.1 | 0.1 | 0.1 | 0.1 |
| Precision | bf16-mixed | bf16-mixed | bf16-mixed | bf16-mixed |
| Batch Size | 128 | 128 | 128 | 128 |
| Max Epochs | 50 | 150 | 100 | 100 |
| Gradient Clipping | 1.0 | 1.0 | 1.0 | 1.0 |
| *Two-Timescale Learning Rate* | | | | |
| Policy LR Scale | 0.25 | 0.25 | 0.25 | 0.25 |
| Transition LR Scale | 1.0 | 1.0 | 1.0 | 1.0 |
| *Architecture* | | | | |
| State Encoder | CNN | CNN | CNN | CNN |
| State Decoder | CNN | CNN | CNN | CNN |
| Policy Network | FiLM-MLP | FiLM-MLP | FiLM-MLP | FiLM-MLP |
| Transition Network | FiLM-MLP | FiLM-MLP | FiLM-MLP | FiLM-MLP |
| Hidden Dimension ($d_{\text{model}}$) | 32 | 32 | 32 | 32 |
| Feedforward Dimension ($d_{\text{ff}}$) | 128 | 128 | 128 | 128 |
| State Dimension | 32 | 32 | 32 | 32 |
| Action Dimension | 16 | 16 | 16 | 16 |
| Num Action Tokens | 1 | 1 | 1 | 1 |
| Max Transition Length (K) | 4 | 1 | 1 | 1 |
| *Latent Action Space* | | | | |
| Action Space Type | Discrete (VQ) | Discrete (VQ) | Continuous (VAE) | Continuous (VAE) |
| Codebook Size | 6 | 36 | – | – |
| Commitment Cost ($\beta$) | 0.25 | 0.25 | – | – |
| Gumbel-Softmax $\tau_{\text{start}}$ | 0.3 | 0.3 | – | – |
| Gumbel-Softmax $\tau_{\text{end}}$ | 0.1 | 0.1 | – | – |
| VAE $\beta$ | – | – | 0.01 | 0.01 |
| *Loss Weights* | | | | |
| Reconstruction Loss | 1.0 | 1.0 | 1.0 | 1.0 |
| Grounding Loss | 0.1 | – | – | – |
| Action VQ Loss | 1.0 | 1.0 | – | – |
| $\lambda_{\text{MDL}}$ | 0.95 | – | – | – |
| *Test-Time Optimization* | | | | |
| Gradient Search Steps | – | – | – | 5 |

*Table 21.* GridWorld $\alpha = 1.00$ Experiments Hyperparameters. We conducted experiments on three seed values.

| Hyperparameter | NEO | Disc-Mono | Cont-Mono | Cont-Mono-Opt |
|---|---|---|---|---|
| *Training* | | | | |
| Learning Rate | $5e{-}4$ | $5e{-}3$ | $5e{-}3$ | $5e{-}3$ |
| Weight Decay | $1e{-}2$ | $1e{-}2$ | $1e{-}2$ | $1e{-}2$ |
| Warmup Ratio | 0.05 | 0.05 | 0.05 | 0.05 |
| LR Scheduler | Cosine | Cosine | Cosine | Cosine |
| Min LR Ratio | 0.1 | 0.1 | 0.1 | 0.1 |
| Precision | bf16-mixed | bf16-mixed | bf16-mixed | bf16-mixed |
| Batch Size | 128 | 128 | 128 | 128 |
| Max Epochs | 50 | 150 | 100 | 100 |
| Gradient Clipping | 1.0 | 1.0 | 1.0 | 1.0 |
| *Two-Timescale Learning Rate* | | | | |
| Policy LR Scale | 0.25 | 0.25 | 0.25 | 0.25 |
| Transition LR Scale | 1.0 | 1.0 | 1.0 | 1.0 |
| *Architecture* | | | | |
| State Encoder | CNN | CNN | CNN | CNN |
| State Decoder | CNN | CNN | CNN | CNN |
| Policy Network | FiLM-MLP | FiLM-MLP | FiLM-MLP | FiLM-MLP |
| Transition Network | FiLM-MLP | FiLM-MLP | FiLM-MLP | FiLM-MLP |
| Hidden Dimension ($d_{\text{model}}$) | 32 | 32 | 32 | 32 |
| Feedforward Dimension ($d_{\text{ff}}$) | 128 | 128 | 128 | 128 |
| State Dimension | 32 | 32 | 32 | 32 |
| Action Dimension | 16 | 16 | 16 | 16 |
| Num Action Tokens | 1 | 1 | 1 | 1 |
| Max Transition Length (K) | 4 | 1 | 1 | 1 |
| *Latent Action Space* | | | | |
| Action Space Type | Discrete (VQ) | Discrete (VQ) | Continuous (VAE) | Continuous (VAE) |
| Codebook Size | 6 | 36 | – | – |
| Commitment Cost ($\beta$) | 0.25 | 0.25 | – | – |
| Gumbel-Softmax $\tau_{\text{start}}$ | 0.3 | 0.3 | – | – |
| Gumbel-Softmax $\tau_{\text{end}}$ | 0.1 | 0.1 | – | – |
| VAE $\beta$ | – | – | 1e-3 | 1e-3 |
| *Loss Weights* | | | | |
| Reconstruction Loss | 1.0 | 1.0 | 1.0 | 1.0 |
| Grounding Loss | 0.1 | – | – | – |
| Action VQ Loss | 1.0 | 1.0 | – | – |
| $\lambda_{\text{MDL}}$ | 1.00 | – | – | – |
| *Test-Time Optimization* | | | | |
| Gradient Search Steps | – | – | – | 5 |

## F.2. Arithmetic Reasoning

*Table 22.* Pretraining hyperparameters for the learned state embeddings in arithmetic reasoning experiments. The same pretrained embedding model is used for NEO and all baselines.

| Hyperparameter | Value |
|---|---|
| *Training* | |
| Learning Rate | $3e{-}3$ |
| Weight Decay | $1e{-}2$ |
| LR Scheduler | Cosine Annealing |
| Min LR Ratio | 0.005 |
| Precision | bf16-mixed |
| Batch Size | 512 |
| Max Epochs | 500 |
| Gradient Clipping | 1.0 |
| *Architecture* | |
| State Encoder | Embedding Layer |
| State Decoder | Linear |
| State Embedding Dimension | 8 |
| *VAE* | |
| VAE $\beta$ | $2e{-}5$ |
| *Loss Weights* | |
| Auto Reconstruction Loss | 1.0 |

*Table 23.* Arithmetic Factorization Reasoning Experiments Hyperparameters (all $\alpha$). We conducted experiments on three seed values.

| Hyperparameter | NEO | Disc-Mono | Cont-Mono | Cont-Mono-Opt |
|---|---|---|---|---|
| *Training* | | | | |
| Learning Rate | $1.5e{-}3$ | $1.5e{-}3$ | $1.5e{-}3$ | $1.5e{-}3$ |
| Weight Decay | $1e{-}2$ | $1e{-}2$ | $1e{-}2$ | $1e{-}2$ |
| Warmup Ratio | 0.05 | 0.05 | 0.05 | 0.05 |
| LR Scheduler | Cosine | Cosine | Cosine | Cosine |
| Min LR Ratio | 0.1 | 0.1 | 0.1 | 0.1 |
| Precision | bf16-mixed | bf16-mixed | bf16-mixed | bf16-mixed |
| Batch Size | 512 | 512 | 512 | 512 |
| Max Epochs | 200 | 200 | 200 | 200 |
| Gradient Clipping | 1.0 | 1.0 | 1.0 | 1.0 |
| *Two-Timescale Learning Rate* | | | | |
| Policy LR Scale | 0.5 | 0.5 | 0.5 | 0.5 |
| Transition LR Scale | 1.0 | 1.0 | 1.0 | 1.0 |
| *Architecture* | | | | |
| State Encoder | Embedding Layer | Embedding Layer | Embedding Layer | Embedding Layer |
| State Decoder | Linear | Linear | Linear | Linear |
| Policy Network | Transformer | Transformer | Transformer | Transformer |
| Transition Network | Cross-Attention | Cross-Attention | Cross-Attention | Cross-Attention |
| Policy $d_{\text{model}}$ / $d_{\text{ff}}$ | 64 / 256 | 64 / 256 | 64 / 256 | 64 / 256 |
| Policy Heads / Layers | 4 / 6 | 4 / 6 | 4 / 6 | 4 / 6 |
| Transition $d_{\text{model}}$ / $d_{\text{ff}}$ | 32 / 128 | 32 / 128 | 32 / 128 | 32 / 128 |
| Transition Heads / Layers | 2 / 4 | 2 / 4 | 2 / 4 | 2 / 4 |
| State Dimension | 8 | 8 | 8 | 8 |
| Action Dimension | 4 | 4 | 4 | 4 |
| Num State Tokens | 6 | 6 | 6 | 6 |
| Num Action Tokens | 1 | 1 | 1 | 1 |
| Max Transition Length (K) | 3 | 1 | 1 | 1 |
| *Latent Action Space* | | | | |
| Action Space Type | Discrete (VQ) | Discrete (VQ) | Continuous (VAE) | Continuous (VAE) |
| Codebook Size | 16 | 40 | – | – |
| Gumbel-Softmax $\tau_{\text{start}}$ | 0.3 | 0.3 | – | – |
| Gumbel-Softmax $\tau_{\text{end}}$ | 0.05 | 0.05 | – | – |
| $\tau$ Scheduling Ratio | 0.25 | 0.25 | – | – |
| EMA Decay | 0.99 | 0.99 | – | – |
| Orthogonal Reg Weight | 10 | 10 | – | – |
| VAE $\beta$ | – | – | $2e{-}4$ | $2e{-}4$ |
| *Loss Weights* | | | | |
| Reconstruction Loss | 1.0 | 1.0 | 1.0 | 1.0 |
| Grounding Loss | 0.5 | – | – | – |
| Action VQ Loss | 1.0 | 1.0 | – | – |
| $\lambda_{\text{MDL}}$ (start $\rightarrow$ end) | $1.01 \rightarrow 0.99$ | – | – | – |
| MDL Scheduling Ratio | 0.1 | – | – | – |
| *Test-Time Optimization* | | | | |
| Gradient Search Steps | – | – | – | 30 |
| Gradient Steps lr | – | – | – | 0.1 |

## F.3. Image Editing

*Table 24.* Pretraining Hyperparameters (State VAE) for Image Editing Experiments. We use the same pretrained model for NEO and Baselines experiments.

| Hyperparameter | Value |
| --- | --- |
| *Training* | |
| Learning Rate | $5e{-}4$ |
| Weight Decay | $1e{-}2$ |
| LR Scheduler | Cosine Annealing |
| Min LR Ratio | 0.005 |
| Precision | bf16-mixed |
| Batch Size | 512 |
| Max Epochs | 500 |
| Gradient Clipping | 1.0 |
| *Architecture* | |
| State Encoder | CNN |
| State Decoder | CNN |
| Hidden Dimension ($d_{\mathrm{model}}$) | 32 |
| Feedforward Dimension ($d_{\mathrm{ff}}$) | 128 |
| Dropout | 0.1 |
| State Dimension | 256 |
| *VAE* | |
| VAE $\beta$ | $1e{-}4$ |
| *Loss Weights* | |
| Auto Reconstruction Loss | 1.0 |

*Table 25.* Image Editing $\alpha = 1.0$ Experiments Hyperparameters. We conducted experiments on three seed values.

| Hyperparameter | NEO | Disc-Mono | Cont-Mono | Cont-Mono-Opt |
|---|---|---|---|---|
| *Training* | | | | |
| Learning Rate | $1e{-}3$ | $1e{-}3$ | $1e{-}3$ | $1e{-}3$ |
| Weight Decay | $1e{-}2$ | $1e{-}2$ | $1e{-}2$ | $1e{-}2$ |
| Warmup Ratio | 0.05 | 0.05 | 0.05 | 0.05 |
| LR Scheduler | Cosine | Cosine | Cosine | Cosine |
| Min LR Ratio | 0.1 | 0.1 | 0.1 | 0.1 |
| Precision | bf16-mixed | bf16-mixed | bf16-mixed | bf16-mixed |
| Batch Size | 64 | 64 | 64 | 64 |
| Max Epochs | 50 | 50 | 50 | 50 |
| Gradient Clipping | 1.0 | 1.0 | 1.0 | 1.0 |
| *Two-Timescale Learning Rate* | | | | |
| Policy LR Scale | 0.25 | 0.25 | 0.25 | 0.25 |
| Transition LR Scale | 0.5 | 0.5 | 0.5 | 0.5 |
| *Architecture* | | | | |
| State Encoder | CNN | CNN | CNN | CNN |
| State Decoder | CNN | CNN | CNN | CNN |
| Policy Network | FiLM-MLP | FiLM-MLP | FiLM-MLP | FiLM-MLP |
| Transition Network | FiLM-MLP | FiLM-MLP | FiLM-MLP | FiLM-MLP |
| Hidden Dimension ($d_{\text{model}}$) | 32 | 32 | 32 | 32 |
| Feedforward Dimension ($d_{\text{ff}}$) | 128 | 128 | 128 | 128 |
| State Dimension | 256 | 256 | 256 | 256 |
| Action Dimension | 16 | 16 | 16 | 16 |
| Num Action Tokens | 1 | 1 | 1 | 1 |
| Max Transition Length (K) | 3 | 1 | 1 | 1 |
| *Latent Action Space* | | | | |
| Action Space Type | Discrete (VQ) | Discrete (VQ) | Continuous (VAE) | Continuous (VAE) |
| Codebook Size | 16 | 64 | – | – |
| Commitment Cost ($\beta$) | 0.25 | 0.25 | – | – |
| Gumbel-Softmax $\tau_{\text{start}}$ | – | – | – | – |
| Gumbel-Softmax $\tau_{\text{end}}$ | – | – | – | – |
| VAE $\beta$ | – | – | 1e-3 | 1e-3 |
| *Loss Weights* | | | | |
| Reconstruction Loss | 1.0 | 1.0 | 1.0 | 1.0 |
| Grounding Loss | 0.1 | – | – | – |
| Action VQ Loss | 0.5 | 0.5 | – | – |
| $\lambda_{\text{MDL}}$ | 1.01 | – | – | – |
| *Test-Time Optimization* | | | | |
| Gradient Search Steps | – | – | – | 30 |
| Gradient Steps lr | – | – | – | 1.0 |

*Table 26.* Image Editing $\alpha = 0.66$ Experiments Hyperparameters. We conducted experiments on three seed values.

| Hyperparameter | NEO | Disc-Mono | Cont-Mono | Cont-Mono-Opt |
|---|---|---|---|---|
| *Training* | | | | |
| Learning Rate | $1e{-}3$ | $1e{-}3$ | $1e{-}3$ | $1e{-}3$ |
| Weight Decay | $1e{-}2$ | $1e{-}2$ | $1e{-}2$ | $1e{-}2$ |
| Warmup Ratio | 0.05 | 0.05 | 0.05 | 0.05 |
| LR Scheduler | Cosine | Cosine | Cosine | Cosine |
| Min LR Ratio | 0.1 | 0.1 | 0.1 | 0.1 |
| Precision | bf16-mixed | bf16-mixed | bf16-mixed | bf16-mixed |
| Batch Size | 64 | 64 | 64 | 64 |
| Max Epochs | 50 | 50 | 50 | 50 |
| Gradient Clipping | 1.0 | 1.0 | 1.0 | 1.0 |
| *Two-Timescale Learning Rate* | | | | |
| Policy LR Scale | 0.25 | 0.25 | 0.25 | 0.25 |
| Transition LR Scale | 0.5 | 0.5 | 0.5 | 0.5 |
| *Architecture* | | | | |
| State Encoder | CNN | CNN | CNN | CNN |
| State Decoder | CNN | CNN | CNN | CNN |
| Policy Network | FiLM-MLP | FiLM-MLP | FiLM-MLP | FiLM-MLP |
| Transition Network | FiLM-MLP | FiLM-MLP | FiLM-MLP | FiLM-MLP |
| Hidden Dimension ($d_{\text{model}}$) | 32 | 32 | 32 | 32 |
| Feedforward Dimension ($d_{\text{ff}}$) | 128 | 128 | 128 | 128 |
| State Dimension | 256 | 256 | 256 | 256 |
| Action Dimension | 16 | 16 | 16 | 16 |
| Num Action Tokens | 1 | 1 | 1 | 1 |
| Max Transition Length (K) | 3 | 1 | 1 | 1 |
| *Latent Action Space* | | | | |
| Action Space Type | Discrete (VQ) | Discrete (VQ) | Continuous (VAE) | Continuous (VAE) |
| Codebook Size | 16 | 64 | – | – |
| Commitment Cost ($\beta$) | 0.25 | 0.25 | – | – |
| Gumbel-Softmax $\tau_{\text{start}}$ | – | – | – | – |
| Gumbel-Softmax $\tau_{\text{end}}$ | – | – | – | – |
| VAE $\beta$ | – | – | 1e-3 | 1e-3 |
| *Loss Weights* | | | | |
| Reconstruction Loss | 1.0 | 1.0 | 1.0 | 1.0 |
| Grounding Loss | 0.1 | – | – | – |
| Action VQ Loss | 0.5 | 0.5 | – | – |
| use $\lambda_{\text{MDL}}$ scheduling | true | – | – | – |
| $\lambda_{\text{MDL}}^{start}$ | 1.05 | – | – | – |
| $\lambda_{\text{MDL}}^{end}$ | 1.0 | – | – | – |
| *Test-Time Optimization* | | | | |
| Gradient Search Steps | – | – | – | 30 |
| Gradient Steps lr | – | – | – | 1.0 |

*Table 27.* Image Editing $\alpha = 0.33$ Experiments Hyperparameters. We conducted experiments on three seed values.

| Hyperparameter | NEO | Disc-Mono | Cont-Mono | Cont-Mono-Opt |
|---|---|---|---|---|
| *Training* | | | | |
| Learning Rate | $1e-3$ | $1e-3$ | $1e-3$ | $1e-3$ |
| Weight Decay | $1e-2$ | $1e-2$ | $1e-2$ | $1e-2$ |
| Warmup Ratio | 0.05 | 0.05 | 0.05 | 0.05 |
| LR Scheduler | Cosine | Cosine | Cosine | Cosine |
| Min LR Ratio | 0.1 | 0.1 | 0.1 | 0.1 |
| Precision | bf16-mixed | bf16-mixed | bf16-mixed | bf16-mixed |
| Batch Size | 64 | 64 | 64 | 64 |
| Max Epochs | 50 | 50 | 50 | 50 |
| Gradient Clipping | 1.0 | 1.0 | 1.0 | 1.0 |
| *Two-Timescale Learning Rate* | | | | |
| Policy LR Scale | 0.25 | 0.25 | 0.25 | 0.25 |
| Transition LR Scale | 0.5 | 0.5 | 0.5 | 0.5 |
| *Architecture* | | | | |
| State Encoder | CNN | CNN | CNN | CNN |
| State Decoder | CNN | CNN | CNN | CNN |
| Policy Network | FiLM-MLP | FiLM-MLP | FiLM-MLP | FiLM-MLP |
| Transition Network | FiLM-MLP | FiLM-MLP | FiLM-MLP | FiLM-MLP |
| Hidden Dimension ($d_{\text{model}}$) | 32 | 32 | 32 | 32 |
| Feedforward Dimension ($d_{\text{ff}}$) | 128 | 128 | 128 | 128 |
| State Dimension | 256 | 256 | 256 | 256 |
| Action Dimension | 16 | 16 | 16 | 16 |
| Num Action Tokens | 1 | 1 | 1 | 1 |
| Max Transition Length (K) | 3 | 1 | 1 | 1 |
| *Latent Action Space* | | | | |
| Action Space Type | Discrete (VQ) | Discrete (VQ) | Continuous (VAE) | Continuous (VAE) |
| Codebook Size | 16 | 64 | – | – |
| Commitment Cost ($\beta$) | 0.25 | 0.25 | – | – |
| Gumbel-Softmax $\tau_{\text{start}}$ | – | – | – | – |
| Gumbel-Softmax $\tau_{\text{end}}$ | – | – | – | – |
| VAE $\beta$ | – | – | 0.0001 | 0.0001 |
| *Loss Weights* | | | | |
| Reconstruction Loss | 1.0 | 1.0 | 1.0 | 1.0 |
| Grounding Loss | 0.1 | – | – | – |
| Action VQ Loss | 0.5 | 0.5 | – | – |
| use $\lambda_{\text{MDL}}$ scheduling | true | – | – | – |
| $\lambda_{\text{MDL}}^{start}$ | 1.01 | – | – | – |
| $\lambda_{\text{MDL}}^{end}$ | 1.0 | – | – | – |
| *Test-Time Optimization* | | | | |
| Gradient Search Steps | – | – | – | 30 |
| Gradient Steps lr | – | – | – | 1.0 |

# G. Extended Related Works

### G.1. Language of Thought and Cognitive Theory Learning.

The Language of Thought hypothesis (Fodor, 1975) proposes that human cognition operates over compositional symbolic representations. Related ideas appear in Bayesian program learning (Ellis et al., 2016) and cognitive models of concept formation, where hypotheses are represented as programs or rules (Lake et al., 2016; Tenenbaum et al., 2011). These approaches highlight the importance of structured, interpretable representations for generalization and explanation. Our work is inspired by this perspective but departs from symbolic modeling: NEO learns both the vocabulary and semantics of a Language of Thought in a neural and data-driven manner. Programs are latent and continuous in execution, while remaining discrete and compositional in structure, enabling theory induction directly from observation.

### G.2. Neural Program Induction & Program Synthesis

A long line of work studies neural program induction and program synthesis from input–output examples, including Neural Programmer-Interpreter (Reed & de Freitas, 2016), differentiable interpreters (Feser et al., 2017), NTM (Graves et al., 2014), NPI (Reed & de Freitas, 2016), LEAPS (Trivedi et al., 2022), HPRL (Liu et al., 2023), LPN (Macfarlane & Bonnet, 2025) and program synthesis frameworks such as DreamCoder (Ellis et al., 2020). These methods typically assume access to symbolic inputs, explicit domain-specific languages, or task-level supervision that specifies the program space. In contrast, our work addresses program induction directly from raw, non-symbolic observations without predefined grammars or program annotations. Moreover, while prior approaches often focus on solving specific tasks, NEO learns a reusable library of primitives and induces latent programs as explanatory theories, emphasizing compositionality and transfer across phenomena rather than task-specific correctness.

### G.3. ARC-AGI.

ARC-AGI (Chollet, 2019; Chollet et al., 2025; 2026) evaluates abstract reasoning by requiring models to infer algorithmic programs from a small number of demonstration pairs. These benchmarks primarily emphasize reasoning and generalization in low-data regimes, while treating the underlying primitives and representational biases as largely fixed rather than learned. In contrast, our work focuses on discovering such primitives directly from raw observations in an unsupervised manner. Furthermore, whereas ARC-AGI is formulated over explicitly defined tasks with support sets, our framework operates on independent $(x, y)$ transitions and supports both program induction and the learning of a reusable compositional language.

### G.4. Compositional and Systematic Generalization.

Adaptive tokenization (Duggal et al., 2024; 2025) and emergent communication methods (Elberg et al., 2025) study variable-length, compositional representations for images learned without explicit supervision, aiming at efficient representation or communication of individual observations. More broadly, compositional generalization has been explored in neural module networks (Andreas et al., 2017), and benchmarks such as SCAN (Lake & Baroni, 2018), gSCAN (Ruis et al., 2020), showing that explicit compositional structure can improve systematic generalization to novel combinations. However, most existing approaches rely on symbolic inputs, hand-designed primitives, or task formulations with few-shot support sets that share an underlying rule. In contrast, our formulation removes these assumptions by learning variable-length, compositional programs directly from raw observation pairs, allowing explanations to adapt to the complexity of the underlying transformation and enabling compositional structure to emerge from data rather than being imposed by dataset design.

### G.5. Latent Action Models (LAM) & World Models.

A line of work including LAPO (Schmidt & Jiang, 2024), AdaWorld (Gao et al., 2025), LAPA (Ye et al., 2025), and Genie (Bruce et al., 2024) studies latent action models that learn dynamics from observation-only video data by inferring unobserved actions as latent variables. These methods are primarily designed as scalable pretraining mechanisms to recover action representations in the absence of action labels, supporting downstream control or prediction. Relatedly, world models such as RSSM and Dreamer (Hafner et al., 2018; 2019; 2020; 2024) focus on learning compact latent dynamics representations to enable prediction and planning. While these approaches aim to model environment dynamics, our framework targets a different problem setting: given an arbitrary observation pair $(x, y)$, we seek to induce a theory that explains the transformation itself. Consequently, our approach is not restricted to single-step next-frame prediction and instead emphasizes learning reusable primitives over more general transformations.

