# OpenReview forum: "Learning to Theorize the World from Observation"
_ICML.cc/2026/Conference — ICML 2026 spotlight_

### Official Review · Reviewer_A7th · 2026-03-05

**Soundness:** 3
**Presentation:** 3
**Significance:** 3
**Originality:** 3
**Overall Recommendation:** 4
**Confidence:** 3

**Summary:**

This paper introduces Learning-to-Theorize (L2T), a paradigm in which an AI system constructs explicit compositional theories of the world, represented as executable programs, from raw observation pairs, without language, task labels, or program supervision. The authors propose the Neural Language-of-Thought Programmer (NLTP): an encoder maps a source observation to a latent state; a theory programmer selects discrete primitive operations from a VQ-VAE codebook conditioned on the current state and target observation; a shared transition model executes primitives sequentially; and a decoder reconstructs the target. Program length is selected via an MDL-inspired penalty, and a state grounding loss anchors intermediate states to the encoder manifold. The authors introduce the Observation-to-Theory Induction Benchmark (OTIB) across three domains (GridWorld, Arithmetic Factorization Reasoning, CIFAR-10 Image Editing), with alpha-ratings controlling training coverage. NLTP substantially outperforms monolithic latent baselines on both compositional and length OOD transfer, and test-time sampling improves results further.

**Compliance With Llm Reviewing Policy:**

Affirmed.

**Final Justification:**

My final recommendation remains 4 (Weak Accept). The paper presents a clear and original perspective on learning executable theories from observation, and the proposed NLTP framework is a technically coherent instantiation of that idea. I view the main strengths as the well-motivated problem formulation, the transfer-focused OTIB evaluation protocol, the strong GridWorld results, and the informative ablations that clarify the roles of state grounding and the MDL-style simplicity bias. The paper is also clearly written and sufficiently detailed for evaluation.

The main weaknesses remain the limited empirical scope, the difficulty of arithmetic length generalization in the base model, and the fact that the main evaluated setting is target-conditioned theory induction from observed transitions. These points limit the reach of the paper’s broader framing, although they do not undermine the value of the contribution as a proof of concept.

The rebuttal addressed my main concerns sufficiently and reinforced my prior assessment. I found the added arithmetic diagnosis, sequence-level correctness analysis, image-editing identity baseline, pretraining-sensitivity discussion, and computational cost comparison all helpful. I encourage the authors to make the target-conditioned setting more explicit in the final version and to scope broader world-modeling claims accordingly.

**Key Questions For Authors:**

(Q1) Base NLTP achieves 0.019-0.038 length-OOD transfer on Arithmetic Reasoning (Table 2). Can you diagnose whether this failure stems from incorrect primitive discovery, composition ordering errors, or numeric precision in multi-step multiplication? A targeted diagnosis would change how this core limitation is interpreted.

(Q2) Because inference conditions on y (Eq. 4), NLTP explains already-observed transitions. Can the model operate in a mode where y is unavailable at inference time? If not, this scope constraint should be stated explicitly and affects the validity of comparisons to predictive world models.

(Q3) The encoder and decoder use pretrained parameters. How sensitive are the main results to this choice, and does fully end-to-end joint training cause instability or qualitative changes in the learned primitives?

(Q4) In GridWorld and Arithmetic Reasoning, where ground-truth programs are known, can you report a sequence-level program correctness metric (up to symmetries), rather than only primitive alignment and output-space transfer?

(Q5) How does the inference cost of NLTP-S as a function of budget B compare to Cont-Mono-Opt test-time optimization? This would help practitioners understand the practical tradeoff between the two approaches.

**Limitations:**

Not fully addressed. The authors note restricted primitive sets, short program lengths, reconstruction-induced semantics that may not be causal, deterministic execution brittleness, and restriction to synthetic benchmarks. The section does not discuss the arithmetic length-OOD failure, the inference cost of NLTP-S, or the post-hoc nature of goal-conditioned inference. Potential negative societal impacts appear limited given the synthetic scope, but downstream use of the method should avoid overstating claims about causal or human-interpretable understanding. Brief additions on these points would improve the section.

**Strengths And Weaknesses:**

Strengths:
- The ablation study is rigorous. Complete collapse when state grounding is removed (primitiveness 0.002, all transfer 0.000) clearly establishes its necessity, and the overcomplete codebook experiment (|E|=36) rules out concerns about assuming ground-truth codebook size.
- The code-primitive alignment matrices provide strong qualitative evidence that the model recovers ground-truth primitives rather than entangled composites, even when some primitives are never directly observed at low alpha settings.
- The transferability evaluation protocol, which infers a program from one (x, y) pair and applies it to a different input from the same mechanism, cleanly operationalizes the distinction between memorization and theory induction, and is a useful contribution.
- The demonstration that compositional structure enables effective test-time scaling while monolithic baselines remain entirely flat (Figure 6a) is a noteworthy and practically relevant finding.
- The paper is well-written and the appendices provide thorough hyperparameter tables and pseudocode that support reproducibility.

Weaknesses:
- All three domains are synthetic and low-dimensional. There is no evidence the approach extends to naturalistic video, noisy observations, or richer temporal dynamics, which substantially limits the reach of the paper's framing.
- Because inference is conditioned on the target y at test time (Eq. 4), the method targets explaining observed transitions; claims about forward prediction or general world modeling should be scoped accordingly.
- Base NLTP achieves near-zero length-OOD transfer on Arithmetic Reasoning (0.019-0.038, Table 2), and the failure mode is not diagnosed. This weakens the length generalization claim for the base model.
- Transfer is evaluated in observation space only. Where ground-truth programs exist, the paper analyzes primitive alignment but does not report sequence-level program correctness, leaving open "right output, wrong program" cases.
- The MDL implementation (Eq. 5) is a heuristic exponential penalty rather than a principled two-part code. Sensitivity to lambda_MDL is high (Table 3), and scheduling strategies vary substantially across domains without principled guidance.
- The encoder and decoder use pretrained parameters; sensitivity to this choice, and whether fully end-to-end training remains stable, is not evaluated.
- Baselines are limited to monolithic latent models; comparison to other structured or compositional alternatives (e.g., object-centric world models, compositional tokenization methods) would strengthen the empirical case.
- Image Editing L1 results are reported without a trivial or identity baseline for reference, making absolute performance difficult to interpret.

---

> ### Author Rebuttal · Authors · 2026-03-31
>
> We thank Reviewer A7th for the detailed and constructive review.
>
> **[R4-W1] Synthetic domains and scalability.**
>
> We acknowledge the synthetic scope (see [R3-W1]). Unlike prior methods (see [R2-Q2] for details), L2T operates in a harder setting — discovering primitives entirely from raw observations without symbolic supervision. The synthetic domains are designed to enable controlled evaluation under this challenging setup. Scaling to more complex domains is an important next step; as a modest effort, we expanded the Arithmetic primitive set and confirmed that NLTP continues to discover correct primitives (see [R2-Q1]).
>
> **[R4-W2] / [R4-Q2] NLTP as a world model.**
>
> NLTP learns $p_\theta(s^\prime | s, z)$ — analogous to action-conditioned transitions in conventional world models, but where the primitives are learned from observation, not predefined. After training, the transition model can **simulate the effect of any primitive**, enabling NLTP to roll out any primitive sequence from $x$, by searching over the learned primitive space. The $y$ in Eq. 4 serves as a **goal** guiding program search, analogous to goal-conditioned world models.
>
> **[R4-W3] / [R4-Q1] Arithmetic length-OOD failure diagnosis.**
>
> We provide a targeted diagnosis. (1) Primitive discovery succeeds: NLTP recovers the primitives near-perfectly (Figs. 4, 13–15). **(2) Primitive selection error is the main bottleneck:** a wrong primitive selection is irrecoverable, and these errors compound over longer horizons. NLTP-S resolves this via test-time sampling. (3) Numeric precision is not an issue: each step accuracy is 0.99.
>
> **[R4-W4] / [R4-Q4] Sequence-level program correctness.**
>
> We evaluate sequence-level program correctness on GridWorld and Arithmetic ($\alpha = 0.33$). Using the code–primitive alignment matrix (Appendix C.2.1), we map each learned code to its best-matching GT primitive, translate the inferred program into a GT primitive sequence, and symbolically execute it on $x$. A program is **correct** if the result equals $y$, and **wrong** otherwise.
>
> We focus on "wrong program, correct output" cases:
>
> **GridWorld**: rare (≤ 1.3%), all involving boundary-crossing moves where the GT primitive cannot execute but the model unexpectedly produces $y$.
>
> **Arithmetic**: 21.9% — some codes specialize to finer-grained operations (e.g., ×(4/3)), so they are labeled "wrong" despite producing correct, transferable outputs.
>
> We note that sequence-level correctness inherently requires a design choice of how to translate learned codes into GT primitives, which is non-trivial when the model discovers its own vocabulary. Transferability, which evaluates directly in observation space, provides a more natural measure of program quality.
>
> **[R4-W5] MDL implementation.**
>
> We agree that Eq. 5 is a heuristic rather than a formal two-part code, but it effectively controls the complexity–expressivity trade-off, analogous to per-step cost penalties in RL. We acknowledge that $\lambda_{\text{MDL}}$ requires tuning (Table 3). The scheduling strategy is shared across domains: initially favor shorter programs for stable learning, then gradually encourage decomposition.
>
> **[R4-W6] / [R4-Q3] Pretrainining sensitivity.**
>
> Thank you for pointing this out. In our initial experiments, joint optimization was more difficult, and since the focus is not on state representation itself, we followed the common practice of using pretrained models (e.g., Zhou et al., 2025; Ha et al., 2018).
>
> Motivated by this question, we have begun investigating joint training. Preliminary results on GridWorld ($\alpha=0.33$) are encouraging (ID: **0.972** vs 0.911, Comp. OOD: **0.986** vs 0.933, Length OOD: **0.964** vs 0.845). We are extending to other domains and will include full analysis in the revision.
>
> **[R4-W7] Baseline comparisons.**
>
> We appreciate this suggestion. We clarify that NLTP's compositionality lies in sequential program composition, not state representation — object-centric models address spatial decomposition of observations, which is an orthogonal concern.
>
> Inspired by this, we designed multi-token baselines selecting 2 or 4 tokens combinatorially from a codebook. Both achieve high ID accuracy (≥ 0.99 at $\alpha$=0.33) yet zero OOD transfer, demonstrating that combinatorial expressiveness alone is insufficient without sequential execution and MDL-grounded primitives.
>
> **[R4-W8] Identity baseline for Image Editing.**
>
> Thank you for this suggestion. The identity baseline (L1 between $x$ and $y$) yields 0.212 (Comp. OOD) and 0.271 (Length OOD); NLTP achieves 0.09–0.12. We will add this as a reference line in the revised figures.
>
> **[R4-Q5] NLTP-S inference cost.**
> See [R3-W2] for detailed cost analysis.
>
> ---
> References
>
> Ha, D. and Schmidhuber, J. World models, 2018. URL https://arxiv.org/abs/1803.10122.
>
> Zhou, G., Pan, H., LeCun, Y., and Pinto, L. DINO-WM: World models on pre-trained visual features enable zero-shot planning, 2025. URL https://arxiv.org/abs/2411.04983.

---

> > ### Author Rebuttal · Reviewer_A7th · 2026-04-01
> >
> > Thank you for the detailed and constructive rebuttal. My main concerns have been addressed sufficiently, and I am keeping my overall recommendation at 4 (Weak Accept).
> >
> > In particular, I found the added arithmetic failure diagnosis, sequence-level correctness analysis, image-editing identity baseline, preliminary pretraining-sensitivity results, and computational cost comparison all helpful and responsive to my review.
> >
> > I still encourage the authors to make the revision (e.g. camera-ready version) more explicit that the main evaluated setting is target-conditioned explanation of observed transitions, and to scope broader world-modeling claims accordingly. However, this is primarily a matter of framing and positioning rather than a remaining technical objection, so it does not change my overall assessment

---

> > > ### Author Response · Authors · 2026-04-02
> > >
> > > Thank you for the careful review and for confirming that your concerns have been addressed. We appreciate the constructive feedback throughout this process, and will keep your suggestion regarding the scoping of target-conditioned inference in mind when preparing the revision.

---

### Official Review · Reviewer_kHx2 · 2026-03-10

**Soundness:** 3
**Presentation:** 3
**Significance:** 3
**Originality:** 4
**Overall Recommendation:** 5
**Confidence:** 4

**Summary:**

The paper proposes a method called Learning-to-Theorize (L2T) which learns to output a latent executable program, given a training dataset of input-output pairs. Notably, the primitives / symbols of this programming language is also learned, as the symbols in the codebook of VQ-VAE. Results on the proposed benchmark, including gridworld, arithmetic, and image editing show that the L2T outperforms the baselines.

**Compliance With Llm Reviewing Policy:**

Affirmed.

**Final Justification:**

My concerns are addressed. I'm raising the score from Weak Accept (4) to Accept (5).

**Key Questions For Authors:**

- How long does it take to train the models? What about inference time? How do these numbers compare with those of the baselines?
- How many training examples are used to train the models? I tried to find this information in the paper, but I couldn't.
- What prevents the tasks to be more complicated? For example, in arithmetic tasks, why did you not include addition, subtraction, and division?

**Limitations:**

yes

**Strengths And Weaknesses:**

Strengths:
- The symbol grounding problem is very important and challenging, and the paper presents an interesting method to give semantics to symbols.
- The experiments are well-designed. Having image editing result helps distinguish the method from many existing program synthesis methods (in addition to the learned semantics).

Weaknesses:
- Similar to many works in program synthesis / cogsci-inspired AI, the domains used in this paper are very simple. The primitives in the arithmetic is literally just multiplication by 2, 3, 5, and 7.
- I would love to see computational resource analyses as I'm assuming the proposed method is a lot more computationally expensive than the baselines? I also would love to understand how many training examples needed for the proposed method to work.

---

> ### Author Rebuttal · Authors · 2026-03-31
>
> We thank Reviewer kHx2 for the positive evaluation and the recognition of our work's originality and experimental design. We address each concern below.
>
> **[R3-W1 & R3-Q3] Domain simplicity and extension to other operations.**
>
> We agree that the current evaluation domains are relatively simple. As stated in our Limitations section, this work is intended as a proof-of-concept for the L2T framework formulation and the NLTP architecture. Our primary contributions are the L2T formulation and empirical validation of the core idea with the NLTP architecture (also, see [R2-Q2] for novelty). The synthetic domains are chosen to enable controlled and rigorous evaluation of whether the model can discover and compose primitives without any supervision. Scaling to more complex domains with richer compositional structure is a natural and important next step that we plan to pursue.
>
> To address the reviewer's concern, we conducted additional experiments expanding the primitive set to 6 primes (adding ×11 and ×13). While this is still a modest increase, NLTP continues to discover primitives:
>
> **NLTP Accuracy**
> |$\alpha$|In-dist. Self-Ex. / Transf.|Comp. OOD Self-Ex. / Transf.|Length OOD Self-Ex. / Transf.|
> |-|-|-|-|
> |1.00|0.704 / 0.653|--|0.031 / 0.020|
> |0.66|0.725 / 0.662|0.381 / 0.340|0.003 / 0.002|
> |0.33|0.85 / 0.829|0.366 / 0.324| 0.021 / 0.013 |
>
> **NLTP-S (B=1024) Accuracy**
> |$\alpha$|In-dist. Self-Ex. / Transf.|Comp. OOD Self-Ex. / Transf.|Length OOD Self-Ex. / Transf.|
> |-|-|-|-|
> |1.00|0.992 / 0.846|--|0.725 / 0.533|
> |0.66|0.955 / 0.813|0.945 / 0.766|0.651 / 0.473|
> |0.33|0.876 / 0.800|0.693 / 0.555|0.342 / 0.238|
>
> Regarding addition, subtraction, and division: thank you for this interesting suggestion. We plan to explore extending to heterogeneous operation types. One anticipated challenge is that addition/subtraction induces relatively small changes in the representation compared to multiplication, making it harder for the policy and transition model to distinguish and process individual primitives. Scaling to diverse operation types and longer program lengths is the main future direction we will be working on.
>
> **[R3-W2 & R3-Q1] Computational cost analysis.**
>
> We note that theory induction prioritizes the quality of the discovered explanation over inference speed. Nonetheless, efficiency is a practical consideration, and we provide a detailed cost comparison.
>
> - **Disc-Mono / Cont-Mono**: a single forward pass
> - **Cont-Mono-Opt**: a forward pass, followed by iterative gradient-based optimization of the program vector at inference.
> - **NLTP**: K sequential forward passes through the transition network.
> - **NLTP-S**: repeats the K-step rollout B times with different sampled programs, scaling as O(K×B).
>
> Within each domain, all methods share the same pretrained encoder/decoder.
>
> **GridWorld** (NVIDIA RTX 3090, batch size 128):
> |Method|Train (ms/batch)|Total Train Time (s)|Inference (ms/batch)|
> |-|-|-|-|
> |Disc-Mono|49.8|5,835 (150 ep)|28.3|
> |Cont-Mono|50.2|5,882 (150 ep)|26.6|
> |Cont-Mono-Opt|—|—|91.2|
> |NLTP (Ours)|75.4|5,890 (100 ep)|41.1|
> |NLTP-S (B=8)|—|—|42.5|
> |NLTP-S (B=64)|—|—|174.0|
>
> **Arithmetic Factorization Reasoning** (NVIDIA RTX 4090, batch size 512):
> |Method|Train (ms/batch)|Total Train Time (s)|Inference (ms/batch)|
> |-|-|-|-|
> |Disc-Mono | 25.0 | 9,920 (200 ep) | 13.0 |
> |Cont-Mono | 23.1 | 9,900 (200 ep) | 12.1 |
> |Cont-Mono-Opt | — | — | 178.9 |
> |NLTP (Ours) | 78.9 | 14,300 (200 ep) | 27.5 |
> |NLTP-S (B=8) | — | — | 42.1 |
> |NLTP-S (B=1024)| — | — | 4867.3 |
>
> **Image Editing** (NVIDIA RTX 4090, batch size 64):
> |Method|Train (ms/batch)|Total Train Time (s)|Inference (ms/batch)|
> |-|-|-|-|
> |Disc-Mono| 58.7 | 26,816 (50 ep) | 24.53 |
> |Cont-Mono| 50.1 | 22,925 (50 ep) | 23.41 |
> |Cont-Mono-Opt| — | — | 571.35 |
> |NLTP (Ours)| 106.7 | 48,779s (50 ep) | 38.06 |
>
> NLTP's training cost is roughly 2× that of baselines due to sequential transitions, and inference is slower than single-pass baselines but faster than Cont-Mono-Opt. For NLTP-S, the additional cost from repeated sampling is a direct trade-off for the significant gains in OOD generalization.
>
> **[R3-Q2] Number of training examples.**
> Thank you for pointing this out. We provide the training and evaluation dataset sizes and will include this information.
>
> **GridWorld:**
> |Setting|Train|ID Test|Comp. OOD Test|Length OOD Test|
> |-|-|-|-|-|
> |α=1.00|100,000|10,000|—|20,000|
> |α=0.66|100,000|10,000|10,000|20,000|
> |α=0.33|100,000|10,000|10,000|20,000|
>
> **Arithmetic Factorization Reasoning:**
> |Setting|Train|ID Test|Comp. OOD Test|Length OOD Test|
> |-|-|-|-|-|
> |α=1.00|279,611|27,961|—|15,317|
> |α=0.66|206,520|20,651|80,401|15,317|
> |α=0.33|147,968|14,796|146,306|15,317|
>
> **Image Editing:**
> |Setting|Train|ID Test|Comp. OOD Test|Length OOD Test|
> |-|-|-|-|-|
> |α=1.00|1,170,000|234,000|—|393,886|
> |α=0.66|1,120,000|224,000|440,000|393,886|
> |α=0.33|720,000|144,000|840,000|393,886|

---

> > ### Author Rebuttal · Reviewer_kHx2 · 2026-03-31
> >
> > My concerns are addressed. I'm raising the score from Weak Accept (4) to Accept (5).

---

> > > ### Author Response · Authors · 2026-04-02
> > >
> > > We sincerely thank you for the thorough evaluation and for acknowledging our responses. We are glad that the additional experiments and analyses addressed your concerns.

---

### Official Review · Reviewer_EE4x · 2026-03-13

**Soundness:** 4
**Presentation:** 4
**Significance:** 3
**Originality:** 2
**Overall Recommendation:** 5
**Confidence:** 5

**Summary:**

This paper introduces a neuro-symbolic framework, modeling latent program induction as theorizing. The authors introduce the Neural Language-of-Thought Programmer (NLTP) model to infer a sequence of discrete latent primitives, execute the primitives via a shared state transition model, and explicitly use a Bayesian Occam’s razor (or minimum description length) objective to prefer shorter programs. Evaluations on the authors’ proposed Observation-to-Theory Induction Benchmark (OTIB), and several toy benchmarks (i.e., GridWorld, Arithmetic Refactorization, Image Editing), show the validity of the proposed method and its strong generalizability.

**Compliance With Llm Reviewing Policy:**

Affirmed.

**Final Justification:**

I recommend acceptance for this paper. Based on the current ratings, I guess the paper can already be accepted or selected as an oral, so I would keep my ratings as the same.

**Key Questions For Authors:**

How much can the model discover more complex abstractions? I would also like a clearer discussion of what exactly is new/gained by the L2T framing over traditional latent program induction?

**Limitations:**

yes

**Strengths And Weaknesses:**

Strengths:

- Overall, I like this paper. This paper has a strong idea and a strong perceptual core. The idea of doing concept induction as theorizing the world is not new, but this paper nonetheless proposes a valid framework for implementing this idea in a clean and concrete way. The paper is also well-written and clearly cognitively inspired in writing style.

- The method is technically good. The VQ-VAE formulation, the MDL penalty, and the transition models are neat and clearly designed.

- The performance is good. On GridWorld and some synthetic tasks, the model shows good length generalization. The transferring examples are also interesting.

Weaknesses:

- The main weakness is that the evaluations or tasks are too toy-ish compared to other theory coding or concept induction tasks. The authors state that ARC is not IID so they designed a new task. But actually, if you view ARC as some kind of meta-learning episodes, then each episode, containing support examples and a query, itself is IID. So it's still a (meta) concept induction task. Some potentially harder domains might be ARC/Atari/BabyAI.

Ref:
- Ahmed, Z., Tenenbaum, J. B., Bates, C., & Gershman, S. J. Synthesizing world models for bilevel planning. Transactions on Machine Learning Research.

- Tsividis, P. A., Loula, J., Burga, J., Foss, N., Campero, A., Pouncy, T., ... & Tenenbaum, J. B. (2021). Human-level reinforcement learning through theory-based modeling, exploration, and planning. arXiv preprint arXiv:2107.12544.

---

> ### Author Rebuttal · Authors · 2026-03-31
>
> We sincerely thank Reviewer EE4x for the thorough and positive evaluation. We are grateful that the reviewer found our framework's formulation clean and its performance compelling. Below, we address the two main concerns raised in the review.
>
> **[R2-Q2] Novelty of Learn-to-Theorize Framing.**
> This is perhaps the most important distinction we wish to clarify. We argue that L2T should be viewed not as an incremental extension of prior program induction methods, but as a fundamentally different approach to a related problem. Most existing approaches rely on one or more of the following strong assumptions:
> - **Manually designed state representations.** Methods such as DreamCoder, NPI, TBRL (e.g., Tsividis et al.) assume access to symbolic or logical state representations that are given a priori. This is a very strong assumption — how to obtain such structured representations from raw perceptual input remains an open challenge without a clear solution in the community. L2T is an attempt to bridge this gap by operating directly on raw pixel-level observations.
> - **Pre-defined programmatic primitives.** These methods use human-designed program spaces (e.g., Python syntax, PDDL, or other domain-specific languages) as the theory representation, where compositional primitives are already given. The model only needs to learn how to compose them. In L2T, no such vocabulary is provided — the model must discover the primitives themselves and learn their semantics from observation alone.
> - **Outsourcing composition to pretrained models.** LLM-based program synthesis (e.g., Ahmed et al.) further outsources the composition problem to large language models, leveraging their extensive pretraining knowledge to search over pre-existing symbolic spaces. This limits applicability to domains where such representations are already available — for instance, in domains such as image editing, where established symbolic representations are not readily available, these methods cannot be directly applied.
>
> These three components — symbolic state representations, predefined primitive vocabularies, and pretrained compositional engines — constitute the core pillars of prior program induction methods. L2T provides a fundamentally different approach by removing all three: it is fully learning-based and fully neural, jointly discovering reusable primitives, learning their operational semantics, and learning how to compose them — all from raw observation pairs without any symbolic supervision, predefined program spaces, or pretrained priors. Removing these assumptions makes the problem significantly more challenging, as the model receives far less structured information and must simultaneously solve discovery, grounding, and composition. L2T represents a first step toward addressing this challenging setting.
>
> **[R2-W1] Relationship to ARC-AGI.**
> key distinction is that ARC-AGI relies on grouped support pairs sharing the same underlying program within each episode — a strong assumption that provides substantial information for identifying the transformation rule. We appreciate the reviewer's point that ARC can be viewed as i.i.d. at the meta-level, but even under this view, each episode still provides far more information than the individual i.i.d. $(x,y)$ pairs that L2T operates on. L2T removes this assumption entirely: the model must discover primitives and infer their compositions from ungrouped, individual observation pairs alone.
>
> **[R2-W1 & R2-Q1] Domain complexity and question about complex abstractions.**
> We appreciate this thoughtful concern, and we agree that the current domains are synthetic and relatively simple. This work is best viewed as a proof-of-concept — its primary contribution lies in the L2T formulation itself, demonstrating for the first time that a fully neural model can discover and compose reusable primitives from observation alone. Scaling to environments with greater visual complexity, longer horizons, stochastic dynamics, and partial observability is an important next step that we are actively pursuing. As an initial step in this direction, we have conducted additional experiments expanding the Arithmetic domain with more primitives, confirming that NLTP continues to discover correct primitives as the compositional space grows (see [R3-W1] below).
>
>
> **[R2-R1 & R2-R2] References.**
>
> Thank you for these relevant pointers. We will incorporate them in the related work section, situating them within the landscape described above.

---

> > ### Author Rebuttal · Reviewer_EE4x · 2026-04-04
> >
> > Thanks for the authors' responses. My concerns have been addressed. I recommend acceptance for this paper. Based on the current ratings, I guess the paper can already be accepted or selected as an oral, so I would keep my ratings as the same.

---

> > > ### Author Response · Authors · 2026-04-06
> > >
> > > Thank you for your feedback and support. We are glad our responses addressed your concerns and appreciate your recommendation.

---

### Official Review · Reviewer_Vym4 · 2026-03-13

**Soundness:** 4
**Presentation:** 4
**Significance:** 4
**Originality:** 4
**Overall Recommendation:** 6
**Confidence:** 4

**Summary:**

This work presents Learning-to-Theorize (L2T), a learning paradigm in which a model learns to predict sequences of actions to translate an input into an output. Crucially, the model learns both the codebook of possible primitive actions as well as their arbitrary composition (in both order and length from unsupervised data alone. The authors contribution is threefold: The authors first theoretically formulate L2T and motivate slight modifications for practical implementation as well as additional loss terms. The authors then propose a benchmark, OTIB, to test their approach in 3 domains (GridWorld, Arithmetic Reasoning, Image Editing). Finally, the authors thoroughly benchmark L2T on OTIB, demonstrating its impressive ability to generalize to novel input-output pairs, and systematically examine its success.

**Compliance With Llm Reviewing Policy:**

Affirmed.

**Final Justification:**

I believe this work to be very strong and relevant for the community, and continue to strongly advocate for its acceptance.

**Key Questions For Authors:**

Please see weaknesses above.

**Limitations:**

yes

**Strengths And Weaknesses:**

# Strengths
1. The paper studies an important problem (learning to generalize systematically by constructing composable programs from unsupervised data) in a novel way, achieving strong impressive results.
2. The motivation of the proposed framework, including its design choices for implementation and the design of additional objectives is very well motivated, theoretically sound, and presented in a clean and easy-to-follow manner (including clear and helpful visualizations).
3. The experimental setup is sound and the proposed method achieves convincing results. The following analysis, including clear metrics, comprehensive ablation and sensitivity studies, and illustrative experiments, demonstrates clearly the appeal of the method.
4. The authors cleanly document hyperparameters, most implementation details, and experiment settings, which should ease reproducibility.

# Weaknesses
1. I'm missing an explanation of how the encoder/decoders were pretrained for each task, or whether these are off-the-shelf pretrained models. Generally, I'm curious whether the authors plan to release their code to ease reproducibility of the results.
2. Self-explainability is never defined, is it simply $d_{\mathrm{obs}}(D_\theta(f_\tau(x^{(1)})), y^{(1)})$, similar to transferability? Also, what metric is used for $d_{\mathrm{obs}}$?

# Minor Issues / Comments
- Figs. 1 & 2 are never mentioned in the text
- Sec. 5.6 / Fig. 4: primitiveness is explained and defined in the appendix, but this isn't linked here
- Sec. 5.6, 3rd paragraph: intervenes -> intervene

---

> ### Author Rebuttal · Authors · 2026-03-31
>
> We sincerely thank Reviewer Vym4 for the thorough and encouraging review. We are grateful for the recognition of our problem formulation, theoretical motivation, experimental rigor, and reproducibility. We address each point below.
>
> **[R1-W1] Encoder/decoder pretraining & code release.**
>
> The encoder and decoder are trained from scratch on each domain's observation space as a standard β-VAE, rather than using any off-the-shelf pretrained model. Specifically, we train a domain-specific $\beta$-VAE to reconstruct observations via an encode-decode cycle, then freeze the encoder and decoder weights during NLTP training. Full architectural details and hyperparameters are provided in Appendix F (Tables 12, 16, 18). We will clarify this procedure more explicitly in the revised manuscript.
>
> We also confirm that we plan to release all code and pretrained checkpoints upon acceptance.
>
> Additionally, we have begun investigating end-to-end joint training (without pretraining) on GridWorld, which we discuss in detail in our response to Reviewer A7th [R4-W6/Q3]. Preliminary results are encouraging, though joint optimization introduces additional engineering challenge that require careful tuning. We are extending these experiments to other domains and will include a thorough analysis in the revised manuscript.
>
> **[R1-W2] Self-explainability definition.**
> Thank you for pointing this out. Self-explainability measures whether the model can reconstruct the target observation from a given phenomenon $(x^{(1)}, y^{(1)})$ using its own inferred program. Formally, the model infers a program $\hat{\tau}$ from $(x^{(1)}, y^{(1)})$ and evaluates $d_{\text{obs}}(D_\theta(f_{\hat{\tau}}(x^{(1)})), y^{(1)})$. This contrasts with transferability, where the program $\hat{\tau}$ inferred from $(x^{(1)}, y^{(1)})$ is applied to a different input $x^{(2)}$ from the same underlying mechanism, evaluating $d_{\text{obs}}(D_\theta(f_{\hat{\tau}}(x^{(2)})), y^{(2)})$. For GridWorld and Arithmetic Reasoning, $d_{\text{obs}}$ is exact match accuracy (1 if all entries match, 0 otherwise); for Image Editing, $d_{\text{obs}}$ is L1 pixel distance. We will add this definition explicitly in the revised text.
>
> **[R1-M1~M3] Minor issues.**
> Thank you for catching these. We will add references to Figures 1 & 2 in the text, link the primitiveness definition in Section 5.6 and Figure 4 to the appendix, and fix the typo (intervenes → intervene).

---

> > ### Author Rebuttal · Reviewer_Vym4 · 2026-04-06
> >
> > Thank you for the insightful comments. I am very curious to see the additional experiments and released code upon acceptance. Given that I already strongly recommend acceptance, I am maintaining my score.

---

> > > ### Author Response · Authors · 2026-04-06
> > >
> > > Thank you for your positive feedback and strong support of our work. We appreciate your interest in the additional experiments and code release. We will ensure that both are included upon acceptance to further improve clarity and reproducibility.

---

### Decision · Program_Chairs · 2026-04-30

**Decision:**

Accept (spotlight)

**Comment:**

This work addresses the question "what does it mean to understand the world?", arguing that mere prediction is not sufficient, and that creating theories of how the world works is necessary to qualify a model as having "understanding". These theories must be built purely from raw, non-textual observations, without human supervision. A framework is provided, and a concrete instantiation, the neural language-of-thought programmer (NLTP), is introduced. Theories are represented as executable programs, which are induced and executed by the NLTP. The superior generalization abilities of the NLTP are thoroughly tested.

This work studies some of the core problems of world modeling: Symbol grounding, compositional modeling, and how to learn those directly from raw data. The methodology (VQ-VAE, MDL penalty, transition model) is technically sound and novel (introducing new, potentially more generally useful concepts, like state grounding loss). The approach is tested rigorously, including on OOD cases using both new compositions and different lengths. It is noteworthy that this architecture can benefit from test time scaling, whereas monolithic models do not. The experimental protocol is well-designed, , utilizing illustrative cases, and also well-documented, helping reproducibility.

Overall, a strong paper that merits acceptance.